# Free fatty-acid transport via CD36 drives β-oxidation-mediated hematopoietic stem cell response to infection

Jayna J. Mistry[1,2,5], Charlotte Hellmich[1,3,5], Jamie A. Moore[1], Aisha Jibril[1], Iain Macaulay[2], Mar Moreno-Gonzalez [4], Federica Di Palma[1], Naiara Beraza[4✉], Kristian M. Bowles [1,3✉] & Stuart A. Rushworth [1✉]

Acute infection is known to induce rapid expansion of hematopoietic stem cells (HSCs), but the mechanisms supporting this expansion remain incomplete. Using mouse models, we show that inducible CD36 is required for free fatty acid uptake by HSCs during acute infection, allowing the metabolic transition from glycolysis towards β-oxidation. Mechanistically, high CD36 levels promote FFA uptake, which enables CPT1A to transport fatty acyl chains from the cytosol into the mitochondria. Without CD36-mediated FFA uptake, the HSCs are unable to enter the cell cycle, subsequently enhancing mortality in response to bacterial infection. These findings enhance our understanding of HSC metabolism in the bone marrow microenvironment, which supports the expansion of HSCs during pathogenic challenge.

[1] Norwich Medical School, University of East Anglia, Norwich Research Park, Norwich NR4 7UQ, UK. [2] Earlham Institute, Norwich Research Park, Norwich NR4 7UH, UK. [3] Department of Haematology, Norfolk and Norwich University Hospitals NHS Trust, Colney Lane, Norwich NR4 7UY, UK. [4] Gut Microbes and Health Institute Strategic Programme, Quadram Institute, Norwich, UK. [5] These authors contributed equally: Jayna J. Mistry, Charlotte Hellmich. ✉email: naiara.beraza@quadram.ac.uk; k.bowles@uea.ac.uk; s.rushworth@uea.ac.uk

The maintenance of hematopoiesis is reliant on the lifelong self-renewal and differentiation of the hematopoietic stem cells (HSCs) into all lineages of mature blood cells[1,2]. Hematopoiesis is a dynamic balance between the opposing cell fates of self-renewal and initiation of hematopoietic differentiation. The HSCs, while predominantly quiescent, rapidly enter the cell cycle in response to infection, this rapid expansion of white blood cells in response to pathogenic stress underpins the mammalian response to infection. At present, the mechanisms by which HSC metabolism is regulated in response to the challenges of pathogenic stimuli are not fully understood.

The bone marrow (BM) microenvironment regulates the production of both hematopoietic and non-hematopoietic cells for the maintenance of blood production under normal and stressed conditions[3,4]. HSC metabolism is finely balanced between glycolysis and oxidative phosphorylation (OXPHOS), to maintain the intrinsic needs of the cell within the constraints imposed by the microenvironment[5,6]. Under "steady-state" conditions HSCs reside in a hypoxic niche where quiescent HSC have low mitochondrial activity and favor anaerobic glycolysis to generate the energy requirements for cell maintenance[5,7,8]. After chemotoxic or pathogenic stimulation HSCs move out-of-quiescence and rapidly switch their metabolic profile towards an increase in mitochondrial activity and OXPHOS-dependent ATP generation[6,9,10]. This switch allows differentiating cells to meet their altered and higher metabolic energy demands associated with expansion and differentiation.

Fatty-acid oxidation is utilized as an energy source by both primitive HSCs and more-committed progenitors to aid self-renewal and differentiation[11]. Moreover, inhibiting fatty-acid oxidation has been shown to reduce stem cell capability. In a malignant setting, the leukemic stem cell (LSC) has been shown to interact with its microenvironment resulting in lipolysis of marrow adipose tissue (MAT), fueling the LSC via fatty-acid oxidation. Specifically, the LSCs expressing the fatty-acid transporter CD36 demonstrated high levels of fatty-acid oxidation, providing those LSC with a survival advantage[12]. In addition, we have also shown that free fatty acids are acquired by acute myeloid leukemia (AML) blasts to enhance their proliferation in vitro and in vivo through a mechanism that increased β oxidation[12,13]. This leads us to hypothesize that the β-oxidation-dependent metabolic switch in leukemia has its "origins" in the physiology of the HSC response to infection.

The majority of invasive non-typhoidal salmonella infections are due to Salmonella Typhimurium (S. typhimurium), and have become a prominent cause of bloodstream infection in African adults and children, associated with 20–25% fatality[14]. We have recently reported that in the context of the challenge of acute S. typhimurium infection, the BM microenvironment drives rapid HSC and leukocyte expansion necessary for host survival, through a process dependent on mitochondria transfer into the HSC from tissue-resident BM stromal cells[15].

Therefore, through studies using S. typhimurium and its outer membrane lipopolysaccharide (LPS) to model acute bacterial infection, we aim to understand if and how the mammalian response to infection involves the acquisition of free fatty acids by HSC in the BM microenvironment. Furthermore, we look to elucidate the mechanisms by which FA transport occurs and how it facilitates the immunometabolic changes required for rapid leukocyte expansion in response to bacterial infection.

## Results

### Infection with S. typhimurium drives long-chain fatty-acid uptake in HSC.
It has previously been shown that serum FFA levels are increased in response to infection[16,17]. Here, we observe increased levels of FFA in the serum of mice treated with S. typhimurium (72 hours) and LPS (16 hours) based on serum IL-6 levels. Moreover, blocking IL-6 inhibits LPS-induced serum FFA levels in vivo (Supplementary 1a–e). In addition, we show that significantly increased cell cycling of HSC in response to LPS and S. typhimurium is occurring at 16 hours and 72 hours post exposure, respectively (Supplementary Fig. 1f, g). To investigate real-time fatty-acid uptake by hematopoietic cells in response to infection we developed an in vivo transplant model in which we transduced CD45.1 lineage negative, CD117-positive cells with firefly luciferase (LK+FF). Adoptive transfer of LK+FF cells into CD45.2 animals was performed (Fig. 1a). Post transplantation animals were injected with the D-luciferin to confirm transduction and transplantation (Fig. 1b). CD45.1 LK+FF engrafted CD45.2-recipient animals were then injected with a probe comprising a luciferin molecule conjugated to a long-chain free fatty acid (FFA-luc) by a cleavable disulfide bond and imaged using bioluminescence. This probe is stable outside of the cell but is reduced by glutathione following lipid uptake into the cell. If there is FFA uptake the luciferin reacts with the luciferase which can be seen by bioluminescent imaging[18]. One-week later, the mice were treated with LPS and 16 hours later we injected the mice with the FFA-luc probe. Live animal imaging confirms activation of luciferase in the BM compartment following LPS treatment, demonstrating in vivo, that long-chain FFA is taken up by hematopoietic cells in response to LPS (Fig. 1c, d). However, it is noted that no background bioluminescence is present in the control group, which we believe is due to the signal threshold not being reached.

Next, we looked to determine which hematopoietic stem and/or progenitor cells (HSPC) have increased lipids during infection. Mice were treated with S. typhimurium for 72 hours then sacrificed (Fig. 1e). Analysis of LSK, MPP, HSC, ST-HSC, and LT-HSC populations (Fig. 1f) showed an increase in intracellular neutral lipid staining at 72 hours compared to control non-infected animals (Fig. 1g). LPS treatment for 16 hours also increased intracellular neutral lipid staining in the HSPC populations (Fig. 1h and supplementary Fig. 1h). To investigate if this was uptake of FFA, LK cells were isolated from the BM of mice infected with 72 hours S. typhimurium or treated with LPS for 16 hours and incubated with a BODIPY-dodecanoic acid fluorescent fatty-acid for 30 minutes (long-chain FFA linked to bodipy). LK cells from S. typhimurium or LPS-treated mice had an increased uptake of FFA when compared to LK cells from untreated animals (Fig. 1i). Fluorescent microscopy images confirm an increase in FFA in the LSK cells from S. typhimurium (72 hours) infected mice (Fig. 1j). Moreover, following 16 hours LPS treatment the LSK, HSC, ST-HSC, and LT-HSC populations had increased uptake of long and short-chain FAs (Supplementary Fig. 2a–c). Together, these experiments show that HSC, MPP, and LSK cells all acquire FFA in the context of bacterial infection.

### Infection increases OXPHOS and dependency on β-oxidation in HSPC.
To understand the metabolic changes occurring in the HSC in response to infection we studied LSK cells isolated from animals infected with S. typhimurium for 72 hours, or treated with LPS for 16 hours. Seahorse metabolic flux analysis measuring oxygen consumption rates (OCR) confirmed increased OXPHOS levels in LSK cells from LPS (16 hours) treated and S. typhimurium (72 hours) infected C57BL/6 J mice (Fig. 2a, b). Extracellular acidification rate (ECAR) was also measured and showed no change in glycolysis following S. typhimurium infection or LPS treatment (Fig. 2c and Supplementary 3a). Cells can utilize glucose, glutamine, and/or fatty acids to generate ATP and

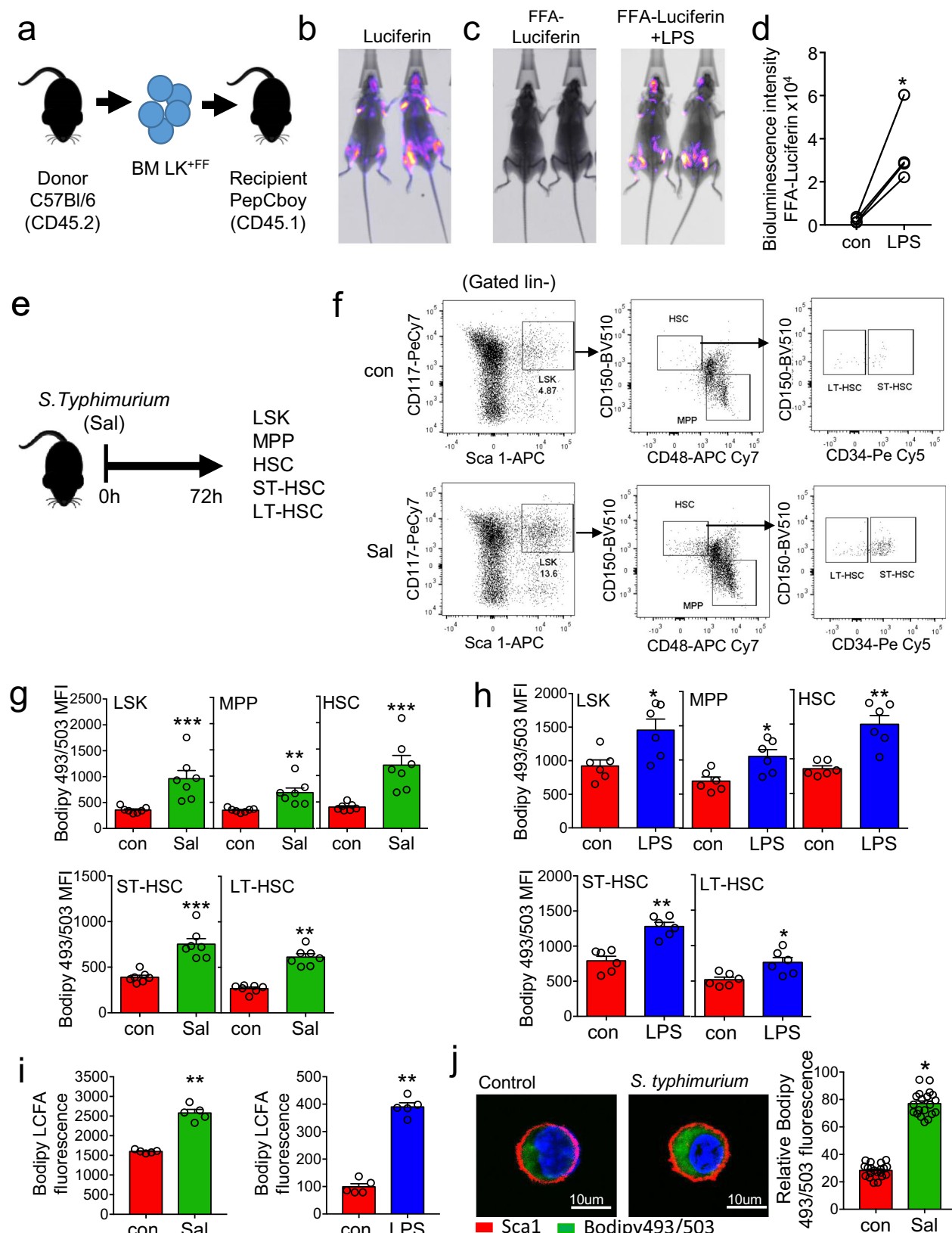

metabolites to support increased cellular activities. We used the Seahorse XF Mito Fuel Flex Test to monitor the dependency on fatty acids as a source of energy in LSK cells from LPS (16 hours) treated animals. Results confirm an increased dependency on fatty-acid oxidation (FAO) compared to control LSK cells (Fig. 2d). To determine the role of β-oxidation in OXPHOS we

treated animals with LPS then isolated LSK cells after 16 hours and treated the cells ex vivo with etomoxir (Eto) a β-oxidation inhibitor specifically an inhibitor of mitochondrial CPT1, an enzyme located on the outer mitochondrial membrane responsible for the catalyzing the first step of FAO. Seahorse XF Mito stress test showed that Eto inhibited the LPS-induced increase in

**Fig. 1 Infection with *S. typhimurium* drives long-chain fatty-acid uptake in HSC. a** Schematic diagram of experimental design. WT CD45.2 lineage negative, CD117-positive (LK) cells were isolated and transduced with firefly luciferase (LK$^{+FF}$) and transplanted into WT CD45.1 animals. **b** Mice were imaged using bioluminescence imaging to confirm engraftment. **c** Mice were injected with control PBS for 16 hours then treated with FFA-SS-luc and imaged using bioluminescence (FFA-luciferin). One-week later mice were injected LPS for 16 hours then treated with FFA-SS-luc and imaged using bioluminescence Representative images of control and LPS-treated mice. **d** Densitometry of the bioluminescent images in (**c**) to determine fluorescence intensity in the vehicle and LPS-treated animals. $n = 4$. **e** Schematic diagram of experiment in which C57BL/6 J mice were infected with *S. typhimurium* (Sal) for 72 hours and analyzed for LSK, MPP, HSC, ST-HSC, and LT-HSC populations by flow cytometry. **f** The gating strategy used to identify the LSK, MPP, HSC ST-HSC, LT-HSC populations are shown. **g** Lipid content (Bodipy 493/503 mean fluorescence intensity (MFI)) was assessed by flow cytometry from control and *S. typhimurium* (Sal) (72 hours) treated mice. $n = 7$ in each group. **h** C57BL/6 J mice were treated with 1 mg/kg LPS for 16 hours, the bone marrow was extracted, and the cells were analyzed by flow cytometry for lipid content (Bodipy 493/503 MFI) $n = 6$ in each group. **i** C57BL/6 J mice were infected with *S. typhimurium* (Sal) for 72 hours or LPS for 16 hours. Long-chain fatty-acid (LCFA) uptake was measured using the QBT assay. $n = 5$ in each group (**j**) C57BL/6 J mice were infected with *S. typhimurium* (Sal) for 72 hours. Representative live-cell fluorescent microscopy images of LSK cells isolated from the mice, Sca1 membrane stain (red), Bodipy 493/503 (green), and Hoechst 33342 (blue). Quantification of Bodipy 493/503 fluorescence in LSK cells from images shown, 20 LSK cells from five mice in each condition. Data shown are means ± SD. The Mann–Whitney *U* test (two-tailed) was used to compare between treatment groups *$p < 0.05$ **$p < 0.01$ ***$p < 0.001$. Source data are provided as a Source Data file.

OCR (both basal and maximal respiration) but had no effect on control LSK cells (Fig. 2e–g).

To understand the significance of β-oxidation in HSC in response to infection we pre-treated mice with Eto followed by LPS (16 hours) or *S. typhimurium* (72 hours) (Fig. 3a). Fig. 3b–e show HSC expansion at 16 hours post LPS treatment or 72 hours post *S. typhimurium* was inhibited by Eto treatment. Treatment with Eto alone did not affect HSC cell cycling (Supplementary Fig. 3b). We next confirmed that CPT1A expression was increased in response to LPS (Fig. 3f). To determine the role of CPT1A in the haematopoietic compartment on cell cycling after induction with LPS, we isolated LK cells from WT mice and transduced them with either control KD (WT $^{(LK\ con\ KD)}$) or CPT1A KD (LK$^{CPT1A\ KD}$) (Supplementary Fig. 3c). Mice received an adoptive transfer of LK$^{CPT1A\ KD)}$ cells (Fig. 3g) and post engraftment, we treated the animals with LPS for 16 hours (Supplementary Fig. 3d). Results confirm that HSC expansion was inhibited in the LK$^{CPT1A\ KD}$ in response to LPS (Fig. 3h–j). These results show that HSC has increased reliance on β-oxidation for their expansion, upon infection-mediated stress stimuli.

**CD36 regulates long-chain free fatty-acid uptake in HSC in response to infection.** Several membrane proteins have been identified to facilitate the trafficking of lipids into and out of cells, including CD206, CD36, fatty-acid-binding proteins (FABPs), and fatty-acid transport proteins (FATPs; also known as solute carrier 27 (SLC27)). The FATP family includes six members (FATP1–FATP6)[19,20]. To determine how FFA are transported into the HSC we assayed the expression of genes known to be involved in lipid uptake[19]. HSC isolated from LPS and *S. typhimurium*-treated animals had consistently higher expression of CD36, Slc27a4, and FABP3 mRNA compared to control HSC isolated from untreated animals, with Slc27a2 not being upregulated at 16 hours post LPS treatment (Fig. 4a). Analysis of CD36, Slc27a4, and FABP3 protein expression showed that CD36 protein was induced on HSC in response to both LPS and *S. typhimurium*. Although, FABP3 protein expression showed a slight increase in response to LPS this was not significant (Supplementary Fig. 4a, b). Moreover, Slc27a4 protein expression was not induced by either LPS or *S. typhimurium* (Fig. 4b and Supplementary Fig. 4a, b). To test the role of CD36 or FABPs in the uptake of FFA we used an inhibitor for CD36 (sulfosuccinimidyl oleate; SSO) and a FABP inhibitor (BMS309403 targets FABP3, FABP4, and FABP5)[21]. Results show that SSO and not BMS309403 inhibited LCFA uptake in HSC (supplementary Fig. 4c). Taken together these data suggested that CD36 is the trafficking protein regulating FFA flux in this context.

To test the functional importance of CD36 in the uptake of FFA and its impact on HSC expansion in response to infection, animals were pre-treated with the CD36 inhibitor SSO before injection with LPS (Fig. 4c). LKs isolated from animals treated with SSO and LPS had reduced uptake of FFA compared with LPS alone (Fig. 4d). Moreover, HSPCs isolated from these animals pre-treated with SSO before stimulation with LPS had a reduced lipid content, lower basal and maximal respiration, and reduced cycling when compared to animals treated with LPS alone (Supplementary Fig. 4d–f and Fig. 4e).

To confirm the effects of pharmacological inhibition of CD36 in response to infection was consistent with the genetic knockout of CD36 we treated WT (CD36$^{+/+}$) and CD36 knockout (CD36$^{-/-}$) animals with LPS for 16 hours (Fig. 4f). Unlike the WT CD36$^{+/+}$ animals, we found LPS-treated CD36$^{-/-}$ mice had no increase in FFA uptake, lipid content, or HSC cycling compared to control CD36$^{-/-}$ mice (Fig. 4g, h and Supplementary Fig. 4g, h). In a similar way, 72 hours after inoculation with *S. typhimurium*, fluorescent microscopy demonstrated no increase in lipid content in the LSK cells from infected CD36$^{-/-}$ mice compared to control CD36$^{-/-}$ mice. (Fig. 4i and Supplementary Fig. 5a).

To understand the impact of CD36 expression on HSC metabolism in response to infection we infected WT CD36$^{+/+}$ and CD36$^{-/-}$ animals with *S. typhimurium* 72 hours. LSK cells were then isolated and analyzed using the Seahorse XF Mito stress test. In contrast to LSK cells from WT CD36$^{+/+}$ mice, we found CD36$^{-/-}$ LSK cells had an increased basal ECAR but no changes in basal OCR in response to *S. typhimurium* infection (Fig. 4j, k and Supplementary Fig. 5b). Maximal OCR respiration was also inhibited in the infected CD36$^{-/-}$ LSK cells compared with infected WT CD36$^{-/-}$ LSK cells (Supplementary Fig. 5b). Moreover, CD36$^{-/-}$ LSK cells from control and LPS-treated mice have reduced dependency on β-oxidation (Supplementary Fig. 5c).

To investigate the role of CD36 on fatty-acid uptake by hematopoietic cells in response to infection we transduced CD36$^{-/-}$ (CD45.2) lineage negative, CD117-positive cells with firefly luciferase (LK$^{+FF}$). These LK$^{+FF}$ cells were then transplanted into WT CD36$^{+/+}$ (CD45.1) animals (Fig. 4l). Animals were then injected with ᴅ-luciferin to confirm transduction and transplantation (Supplementary Fig. 5d). CD36$^{-/-}$ LK$^{+FF}$ engrafted CD45.1 recipient animals were then injected with luciferin conjugated long-chain free fatty-acid (FFA-luc) probe and imaged using bioluminescence. One-week later the mice were treated with LPS and 16 hours later we injected the mice again with the FFA-luc probe. Live animal imaging confirms no uptake of FFA-luc in the BM compartment, demonstrating in vivo, that long-chain FFA is not taken up by CD36$^{-/-}$ hematopoietic cells

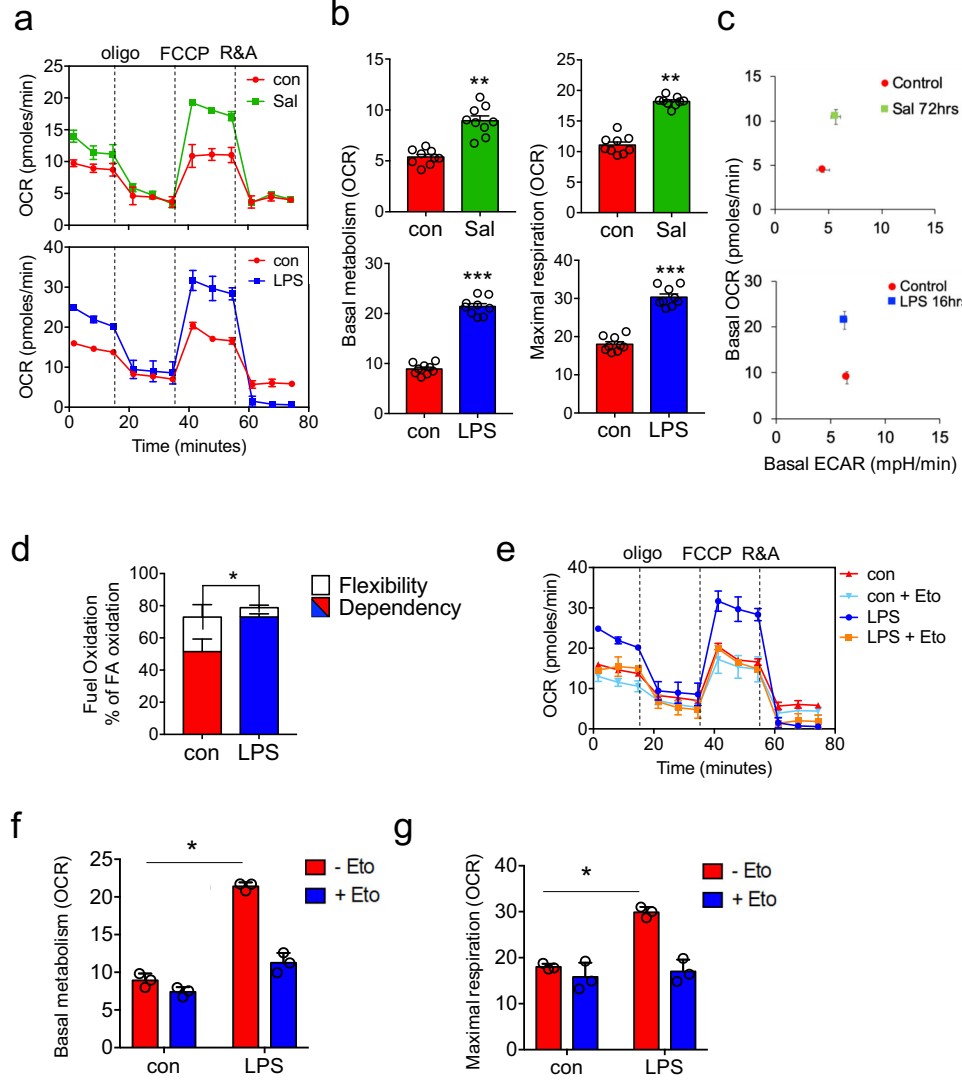

**Fig. 2 Infection increases OXPHOS and dependency on β-oxidation in HSPC. a** C57BL/6 J mice were infected with *S. typhimurium* (Sal) for 72 hours or treated with 1 mg/kg LPS for 16 hours, the animals were killed and the LSK population was isolated by FACS. oxygen consumption rate (OCR) was measured by the extracellular flux assay. Representative seahorse trace from mice, $n = 3$ in each group. **b** Basal (normalized to rotenone) and maximal mitochondrial respiration in LSK cells from control, LPS 16 hours, and *S. typhimurium* (72 hours) treated mice. $n = 6$ in each group. **c** Basal extracellular acidification rate (ECAR) compared with basal OCR levels provides a snapshot of the bioenergetic profile of LSK before and after treatment with LPS (16 hours) or *S. typhimurium* (72 hours). Basal OCR normalized to rotenone. $n = 5$ in each group. **d** C57BL/6 J mice were treated with 1 mg/kg LPS for 16 hours, the LSK population was isolated by FACS. The LSK population was analyzed by seahorse mitochondrial fuel flex kit for the reliance on long-chain fatty acids to maintain baseline respiration. $n = 5$ in each group. **e** C57BL/6 J mice were treated with 1 mg/kg LPS for 16 hours, the animals were killed and the LSK population was isolated by FACS. LSKs were treated with 4 μM of the β-oxidation inhibitor, etomoxir (Eto) for 1 hour, and OCR levels were measured by the extracellular flux assay. $n = 5$ mice in each group. **f** Basal mitochondrial respiration (normalized to rotenone) of LSK cells from control and LPS-treated animals with and without eto. $n = 3$ mice in each group. **g** Maximal mitochondrial respiration LSK cells from control and LPS-treated animals with and without eto. $n = 3$ mice in each group. Data shown are means ± SD. The Mann–Whitney $U$ test (two-tailed) was used to compare between two treatment groups. *$p < 0.05$ **$p < 0.01$ ***$p < 0.001$. Source data are provided as a Source Data file.

in response to LPS (Supplementary Fig. 5e and Fig. 4m). Together, these data show that CD36 on HSCs is essential for the uptake of FFA in response to infection.

**FFA uptake through CD36 is an essential component of HSC expansion in response to infection.** To determine if the uptake of FFA response is required for HSC expansion we isolated LK cells from WT CD36$^{+/+}$ (CD45.1) mice and transplanted them into CD36$^{-/-}$ (CD45.2) animals (Fig. 5a). Therefore, these animals were CD36$^{-/-}$ but had a WT CD36$^{+/+}$ haematopoietic system. Engraftment of CD36$^{+/+}$ (CD45.1) cells are shown in

Supplementary Figs. 6a–e and 7a. We found that CD36 expression was elevated in the HSC from the CD36$^{+/+}$ transplanted into CD36$^{-/-}$ mice treated with LPS (Fig. 5b). Moreover, following LPS treatment we observed increased uptake of FFA, increase in lipid content, and increased expansion of HSCs, similar to the WT animal response (Fig. 5c, d and Supplementary Fig. 7b–d). In addition, LSK cells from transplanted WT CD36$^{+/+}$ (CD45.1) into CD36$^{-/-}$ (CD45.2) animals reversed the metabolic phenotype observed in CD36$^{-/-}$ mice in response to LPS with an increase in OXPHOS, both basal and maximal mitochondrial respiration (Fig. 5e, f).

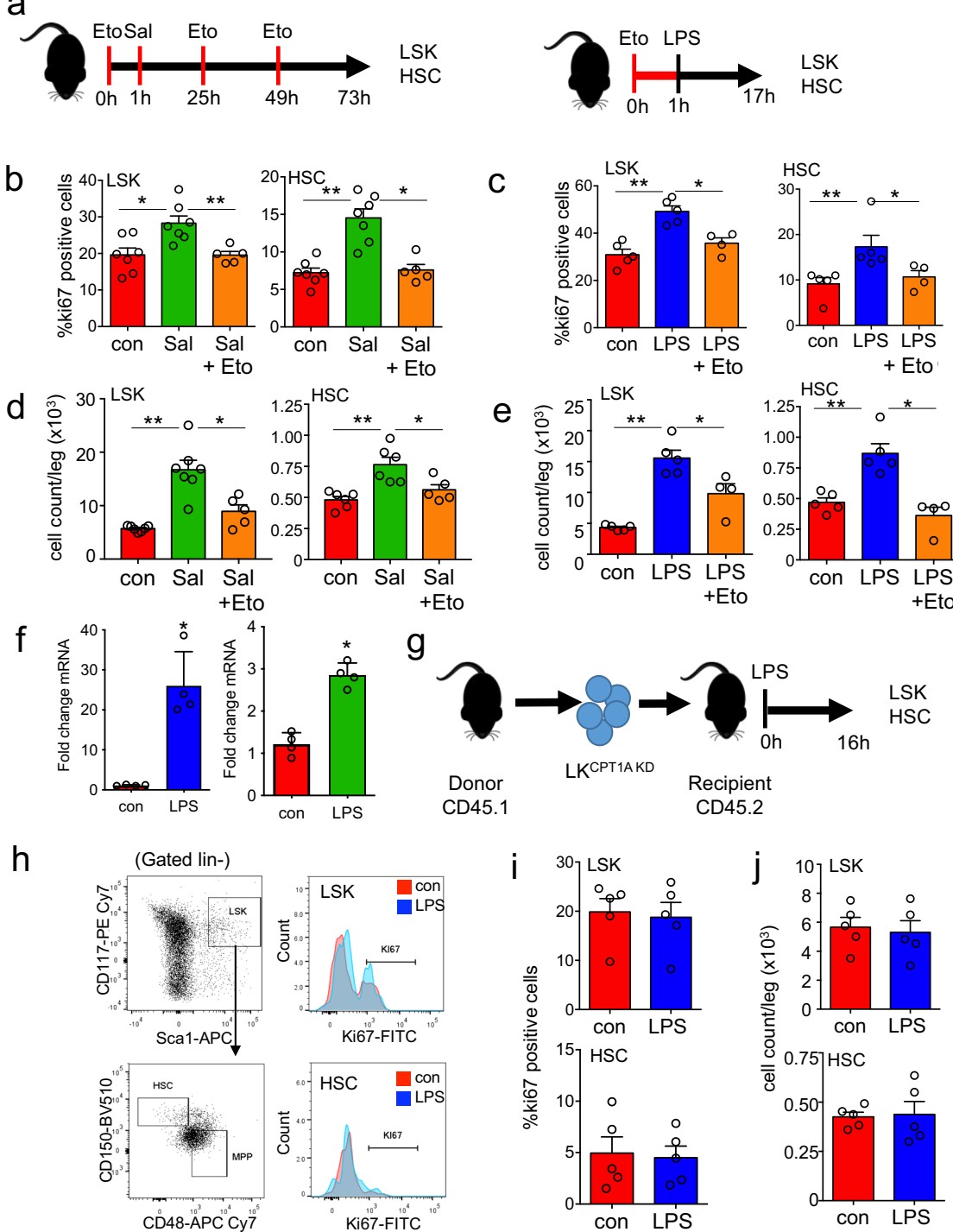

Finally, we wanted to understand the importance of CD36 in the hematopoietic compartment in response to *S. typhimurium* infection. WT (CD45.1) mice received an adoptive transfer of either WT CD36$^{+/+}$ (CD45.2) or CD36$^{-/-}$ (CD45.2) LK cells. These transplanted mice were termed WT$^{(CD36-/-)}$ for CD36 knockout into WT and WT$^{(CD36+/+)}$ for WT CD36$^{+/+}$ into WT. Post engraftment the mice were infected with *S. typhimurium* for 4 days (Fig. 5g and Supplementary Fig. 7e). *S. typhimurium* infected WT$^{(CD36-/-)}$ transplanted animals showed enhanced mortality and weight loss compared with infected WT$^{(CD36+/+)}$ transplanted animals (Fig. 5h, i). Further analysis evidenced increased liver injury

in WT$^{(CD36-/-)}$ compared with WT$^{(CD36+/+)}$ after *S. typhimurium* infection, as shown by the increased alanine aminotransferase (ALT) and aspartate aminotransferase (AST) levels in serum samples and the wide areas of necrosis observed in Hematoxylin & Eosin (H&E)-stained histological liver sections (Fig. 5j, k). We then validated the differentiation ability of CD36$^{-/-}$ HSCs at steady-state and during infection. CD36$^{-/-}$ mice were treated with control PBS (CD36$^{-/-con}$) or LPS (CD36$^{-/-LPS}$) for 16 hours, the mice were sacrificed, the HSCs were isolated and transplanted into WT (CD45.1) mice (Supplementary Fig. 8a). We found no differences in the myeloid/lymphoid ratio in the peripheral blood or the

**Fig. 3 Infection increases dependency on β-oxidation. a** Schematic diagram of experimental design in which C57BL/6 J mice were infected with *S. typhimurium* (Sal) for 72 hours and 10 mg/kg/day Etomoxir (Eto) or 1 mg/kg LPS for 16 hours and 10 mg/kg Eto. The bone marrow was extracted, and the cells were analyzed by flow cytometry for LSK and HSC. **b** Percentage of cycling HSC and LSK as measured by Ki67-positive cells after 72 hours of *S. typhimurium* and 10 mg/kg/day Eto treatment. *n* > 5 mice in each group. **c** Percentage of cycling HSC and LSK as measured by Ki67-positive cells after 16 hours of 1 mg/kg LPS and 10 mg/kg Eto treatment. *n* > 4 mice in each group. **d** Number of LSKs and HSCs leg after 72 hours *S. typhimurium* and Eto treatment. *n* > 5 in each group. **e** Number of LSKs and HSCs per leg after 16 hours of LPS and Eto treatment. *n* = 5 mice in each group. **f** C57BL/6 J mice were treated with 1 mg/kg LPS for 16 hours or *S. typhimurium* for 72 h, HSC were FACS-sorted from control and LPS-treated animals. RNA was analyzed for CPT1A gene expression by qPCR. *n* = 4 in each group. **g** Schematic diagram of experimental design. WT CD45.1 lineage negative, CD117-positive (LK) cells were transduced with a CPT1A knockdown lentivirus (LK^CPT1A KD) were transplanted into WT CD45.2 animals. Post engraftment mice were treated with 1 mg/kg LPS for 16 hours. **h** The bone marrow was extracted, and analyzed by flow cytometry for the LSK and HSC population. **i** Percentage of cycling HSC and LSK as measured by Ki67-positive cells after 16 hours of 1 mg/kg LPS treatment. *n* = 5 mice in each group. **j** Number of LSKs and HSCs per leg after 16 hours of LPS treatment. *n* = 5 mice in each group. Data shown are means ± SD. The Mann–Whitney *U* test (two-tailed) was used to compare between two treatment groups and the Kruskal–Wallis test was followed by Dunn's multiple comparison post hoc test to compare between three treatment groups. *P < 0.05 **p < 0. 01. Source data are provided as a Source Data file.

frequency of differentiated progenitor (CMP GMP or MEP) cells in the BM of CD36$^{-/-}$con or CD36$^{-/-}$LPS transplant mice, 12 weeks post engraftment (Supplementary Figs. 8b, c and 9a–d). Therefore, the defects caused by CD36$^{-/-}$ in HSC may only be evident during stress. Together these data show that CD36 is an important mediator for the HSC response to infection.

## Discussion

Here we report that HSCs actively uptake long-chain FFA in response to acute bacterial infection. This process facilitates an increased reliance, within the HSCs from glycolytic metabolism towards β-oxidation and a proliferation phenotype. FFA uptake occurs after the onset of the transcriptional changes to the fatty-acid transporter CD36. Furthermore, we identified that without surface CD36, HSC is not able to switch from glycolytic metabolism towards β-oxidation and does not enter the cell cycle, leading to a higher susceptibility and increased mortality to infection. Overall, these results provide insights into the metabolic changes in the hematopoietic system which underpin leukocyte expansion and the mammalian response to infection.

The transition from steady-state to emergency hematopoiesis is established to involve a complex remodeling and interplay between hematopoietic and non-hematopoietic cells of the BM microenvironment mediated by cytokines and growth factors[22–24], and this places a significant metabolic demand on the hematopoietic system. Here, we show how FFA uptake by the HSC facilitates the rapid onset of emergency hematopoiesis by facilitating the increase in intracellular fatty acids and subsequent metabolic changes in HSC. These processes are similar to previous observations in AML, whereby, FFAs are acquired by AML blasts to enhance their proliferation both in vitro and in vivo through a mechanism that increases β-oxidation in the tumor cells[13]. Together, the data generated from these studies suggest that AML has hijacked the normal FFA-HSC mechanism in response to stress-induced activation and differentiation. More broadly these data lead us to hypothesize that tumors arising in fat-rich tissues may be dependent on metabolic processes that arise from physiological stress responses in the cell of origin.

FFA uptake is now widely recognized as a fundamental process, underpinning cell metabolism in both malignant and non-malignant tissue. Intracellular FAO and lipid biogenesis pathways are common regulators of stem cell fate. In general, FAO appears fundamental in maintaining the stem cell state, whereas lipid synthesis tends to result in proliferation and differentiation. However, the mechanisms through which these lipid metabolic pathways affect and alter tissue-specific stem cell behavior, in health and disease, appear to differ[25]. As an example, in the non-malignant setting, quiescent HSCs have high fatty-acid oxidation

rates, and disruption of fatty-acid oxidation leads to stem cell differentiation through a mechanism mediated by peroxisome proliferator-activated receptor δ[11]. Our study finds that during infection HSCs take up FFA as a result of upregulation of the fatty-acid translocase CD36 on the cell surface. CD36 belongs to a family of proteins that bind, transport, and take up long-chain fatty acids or function as regulators of these processes[26,27]. In the past several years, the membrane protein CD36 has been extensively studied for its role in facilitating fatty-acid uptake and oxidation and implicated in the pathophysiology of the heart[28] and liver[29] and associated with dysfunctional fatty-acid metabolism[30]. In fact, it is because CD36 is expressed on many different cell types, and also because of the plurality of the disease phenotypes related in some way to CD36 function, that in order to determine the cell-autonomous effects of CD36 function in HSC we decided to develop the model system using knockout CD36 in HSCs and then transplanting these CD36$^{-/-}$ HSC cells back into a wild-type animal, rather than use CD36 knockout animals for our studies. In context, our present study identifies CD36 upregulation in HSC which mediates fatty-acid uptake and allows HSC expansion facilitating the response to infection. It is also important to understand the role of CD36 non-autonomous mechanism. How CD36 is regulated in HSC is not fully known, but others have shown that C/EBPα and C/EBPβ can directly up-regulate CD36 gene transcription through a C/EBP-responding element at the proximal promoter[31]. Moreover, both C/EBPα and C/EBPβ have been shown to mediate steady-state and emergency granulopoiesis respectively[32]. At the epigenetic level, the enhancers and promoters of CD36 are subject to both histone acetylation and methylation[33]. Differentiation of HSC [CD36 negative] into CD36 expressing erythroid precursors is associated with global chromatin modification patterns. Specifically, Cui and colleagues link the H3K4me3 mark at the CD36 promoter to differentiation of HSC to erythroid precursors. Taken together, these studies in the context of the current work suggest that CD36 forms part of a tightly regulated metabolic switch central to the mammalian haematopoietic response to infection.

This study does not ask the question about the source of FFA. However, BM adipose tissue (MAT), is biologically active energy storage and endocrine organ and accounts for ~70% of BM volume in adult humans[34]. Moreover, BM adipocytes are known to increase with age in humans and rodents[35,36]. These adipocytes are not merely passive occupants of the BM but are now appreciated to be actively involved in processes linked to bone metabolism[37], osteoporosis[38], inflammation[39], and regulation of the hematopoietic niche[40]. In addition, BM adipocytes have been shown to support the proliferation of tumors located in the BM including AML[13], multiple myeloma[41], and metastatic solid

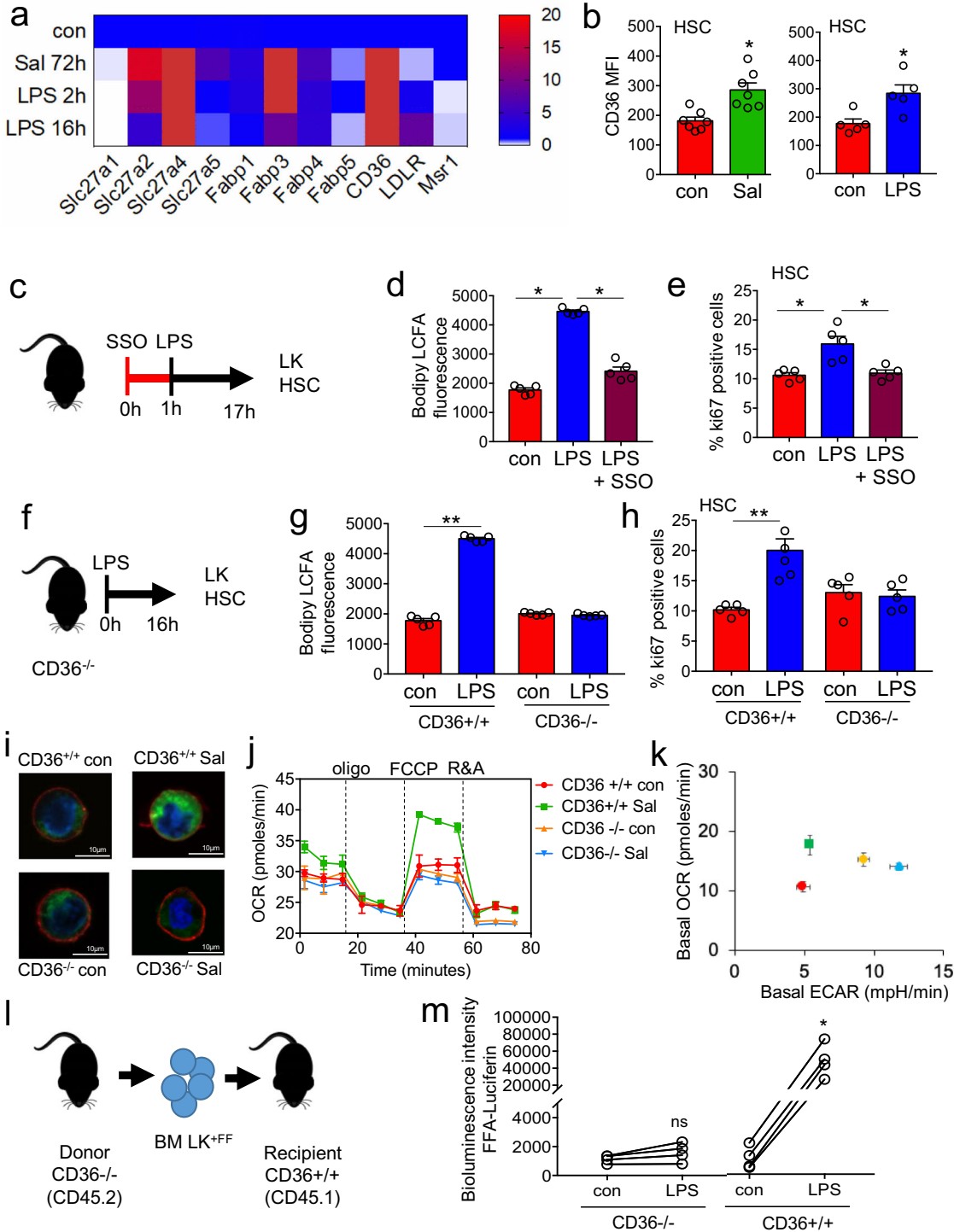

tumors[42]. However, what stimulus is responsible for lipolysis to occur in response to infection is still unknown. There are some data to suggest that endogenous IL-6 is important in regulating both uptake and release of fatty acids from adipocytes[43,44]. Here we show an association between induced IL-6 and FFA uptake in HSC. Moreover, we observed that blocking IL-6 with a monoclonal antibody reduced the serum content of FFA in response to LPS, suggesting that IL-6 is involved in regulating lipolysis in response to infection.

In conclusion, we report that adaptations in FFA uptake and FAO functionally support the hematopoietic response to bacterial infection. Furthermore, in doing so we provide a base for further studies investigating how benign and malignant stem cells in other tissues utilize FFA in response to cellular stress, and specifically whether the provision of FFA by adipocytes in the physiologic response to infection, is altered in individuals with particular vulnerabilities with respect to infection, including older people and those with obesity.

## Methods

**Animals**. C57BL/6 J mice (CD45.2), were purchased (Charles River Massachusetts, United States). B6.SJL-Ptprca[Pep3b/BoyJ] (CD45.1) (PepCboy) mice and B6.129S1-Cd36[tm1Mfe]/J (CD36[−/−]) were purchased from The Jackson Laboratory (Bar Harbour, ME, USA). Mice were individually ventilated and housed under specific pathogen-free conditions in a 12/12-hour light/dark cycle with food and water

**Fig. 4 CD36 regulates long-chain free fatty-acid uptake in HSC in response to infection. a** Heat map of fatty-acid transporter genes differentially expressed by HSCs from control, *S. typhimurium* (Sal), and LPS-treated animals. **b** Flow cytometry analysis of CD36 expression in the HSC of *S. typhimurium* and LPS-treated animals. $n = 5$ in each group. **c** C57BL/6 J mice were pre-treated with SSO for one hour before treatment with LPS for 16 hours. $n = 5$ in each group. **d** The cells from control, LPS, or LPS and SSO-treated animals were isolated and long-chain fatty-acid (LCFA) uptake was measured using the QBT assay. $n = 5$ in each group. **e** Percentage of cycling HSCs as measured by Ki67-positive cells after pre-treatment followed by LPS treatment. $n = 5$. **f** Schematic diagram of the experiment. **g** The LK cells from LPS-treated CD36$^{-/-}$ or WT (CD36$^{+/+}$) mice were isolated and long-chain fatty-acid (LCFA) uptake was measured using the QBT assay. $n = 5$. **h** Percentage of cycling HSCs from LPS-treated CD36$^{-/-}$ or WT (CD36$^{+/+}$) mice as measured by Ki67-positive. $n = 5$ mice in each group. **i** CD36$^{-/-}$ or WT (CD36$^{+/+}$) were infected with *S. typhimurium* (Sal). Representative live-cell fluorescent microscopy images of LSK cells isolated from the mice, Sca1 membrane stain (red), Bodipy 493/503 (green) and Hoechst 33342 (blue), 20 LSK cells from five mice in each condition. **j** CD36$^{+/+}$ and CD36$^{-/-}$ mice were treated with *S. typhimurium* the animals were killed and the LSK population was isolated by FACS, OCR was measured. $n = 5$. **k** Basal ECAR compared to basal OCR levels before and after treatment with LPS or *S. typhimurium*. Basal OCR normalized to rotenone. $n = 5$. **l** Schematic diagram of experimental design. **m** Mice were injected LPS for 16 hours then treated with FFA-SS-luc and imaged using bioluminescence (FFA-luciferin+LPS). Densitometry of the bioluminescent images to determine fluorescence intensity. $n = 4$ for each group. The Mann–Whitney $U$ test (two-tailed) was used and the Kruskal–Wallis test followed by Dunn's multiple comparison post hoc test was used *$P < 0.05$ **$p < 0.01$. Source data are provided as a Source Data file.

provided *ad libitum*. The room temperature for mice was 22 °C and the relative humidity is kept at between 45% and 65%. All animal work used in this study was carried out in accordance with regulations set by the UK Home Office and the Animal Scientific Procedures Act 1986. Mice used were at 8–12 weeks of age and both genders were used for experiments with the exception of mice that were used for transplantation which 3–4-week-old mice were used.

**Cell isolation and preparation**. BM isolation was prepared by isolating the tibia, femur, and pelvis of each mice. The bone was cut in the middle and placed in a 0.5 ml Eppendorf tube in which a hole was made to allow the removal of the BM, placed in an intact 1.5 ml Eppendorf and centrifuged $1000 \times g$ for 6 seconds to collect the BM cells. The BM pellet from each mouse was pooled and washed in PBS, where needed the red cells were lysed using 1× red blood cells lysis buffer (ThermoFisher, Waltham, MA, USA) and centrifuged at $400 \times g$ for 5 min, and the pellet was resuspended in antibody cocktails in PBS.

**BM transplantation**. For the FFA-luciferase allograft mouse model C57BL/6 J and CD36$^{-/-}$ mice expressing the CD45.2 allele antigen were used in the transplant experiments. C57BL/6 J and CD36$^{-/-}$ mice were killed, BM was isolated, lineage depleted followed by CD117 enrichment using CD117+ enrichment kit (LK cells). The LK cells were then seeded at a density of $2 \times 10^5$ cells/well in Dulbecco's Modified Eagle Medium (DMEM) supplemented with 10% FBS plus 1% penstrep with mSCF, mIL3, mIL6 (Peptrotech, NJ, USA). The cells were transduced with pCDH-luciferase-T2A-mCherry virus, which was kindly provided by professor Irmela Jeremias, (Helmholtz Zentrum München, Munich, Germany). Following a 24-hour incubation successful transduction was confirmed by detection of mCherry fluorescence on a fluorescent microscope. The transduced cells were then transplanted into the tail vein of 3–4-week-old PepCboy mice (CD45.1) by intravenous injection which had been preconditioned with busulfan 25 mg/kg/day for 3 days prior to transplantation. FFA-luciferase allograft mouse engraftment was monitored by in vivo bioluminescent live animal imaging.

For the Carnitine Palmitoyltransferase 1A (CPT1A) KD allograft mouse model C57BL/6 J mice expressing the CD45.2 allele antigen was used in the transplant experiments to determine the role of CPT1A in the response to infection. PepCboy mice were killed, BM was isolated, lineage depleted followed by CD117 enrichment using CD117+ enrichment kit (LK cells). The LK cells were then seeded at a density of $2 \times 10^5$ cells/well in DMEM supplemented with 10% FBS plus 1% penstrep with mSCF, mIL3, mIL6 (Peptrotech, NJ, USA). The cells were transduced with MISSION shRNA TRCN0000036279 (human CPT1A shRNA). MISSION pLKO.1-puro Control Vector was used as the lentivirus control (control short hairpin RNA (shRNA)). Viruses were produced as previously described. Plasmids containing MISSION shRNA TRCN0000036279 (human carnitine palmitoyltransferase IA [CPT1A] shRNA) were purchased from Sigma-Aldrich, and viruses were produced as previously described[13]. Lentiviral stocks were concentrated using Amicon Ultra centrifugal filters, and titers were determined using Lenti-Xquantitative real-time polymerase chain reaction (RT-PCR) titration kit (CloneTech). The transduced cells were then transplanted into the tail vein of 3–4-week-old C57BL/6 J mice by intravenous injection which had been preconditioned with 25 mg/kg/day for 3 days prior to transplantation. Successful transduction was assessed using qPCR, if the knockdown was successful the mice were used for further preparations. Post engraftment the animals were treated with 1 mg/kg LPS or control PBS for 16 hours and sacrificed. The BM was extracted and analyzed by flow cytometry for cell cycling and lipid content. The BM was analyzed for CD45.1 expression to determine cell engraftment. If >50% of CD45.1 cells were detected in the BM, the cells were determined to be engrafted.

For the PepCboy CD36 allograft model PepCboy mice expressing the CD45.1 allele antigen were used in this transplant experiment to assess the recovery of the hematopoietic system in CD36 knockout mice in response to infection. PepCboy

mice were sacrificed, and the BM was isolated, lineage depleted followed by CD117 enrichment (LK cells). In all, $2 \times 10^5$ isolated LK PepCboy cells were then injected into the tail vein of 3–4-week-old CD36$^{-/-}$ mice by intravenous injection which had been preconditioned with busulfan for 25 mg/kg/day for 3 days prior to transplantation. Post engraftment the animals were treated with 1 mg/kg LPS or control PBS for 16 hours and sacrificed. The BM was extracted and analyzed by flow cytometry for CD36 expression, lipid uptake, cell cycling, and CD45.1 cell engraftment. The LSK cells were also isolated, and metabolic changes were assessed by seahorse metabolic flux analysis.

For the CD36 PepCboy allograft model PepCboy mice expressing the CD45.1 allele antigen were used in this transplant experiment to assess the role CD36 in the haematopoietic system in response to infection. C57BL/6 J (WT CD36$^{+/+}$) and CD36 knockout (CD36$^{-/-}$) mice were sacrificed, and the BM was isolated, lineage depleted followed by CD117 enrichment (LK cells). LK WT CD36$^{+/+}$ and LK CD36$^{-/-}$ were then injected into the tail vein of 3–4-week-old PepCboy mice by intravenous injection which had been preconditioned with 25 mg/kg/day for 3 days prior to transplantation. These transplanted mice were termed WT$^{(CD36-/-)}$ for CD36 knockout into WT and WT$^{(CD36+/+)}$ for WT into WT. Post engraftment the animals were infected with *S. typhimurium* for 4 days. The BM was extracted and analyzed by flow cytometry.

For all transplantation experiments, engraftment was checked by CD45.1/2 expression on differentiated cells by blood sampling 8–12 weeks post transplantation. Four weeks post engraftment animals were treated.

**FFA-luciferin assay**. Once engrafted the FFA-luciferase allograft mouse model was intraperitoneally injected with 100 µL of 200 µM FFA-SS-luc (SwissLumix Sarl, Switzerland) (0.014 mg/mouse) bound to 0.1% (w/v) bovine serum albumin (BSA) in PBS (a pentadecanoic acid, 16 carbon long-chain fatty-acid 15-carboxyl pentadecyl) immediately prior to imaging. Luminescent images were acquired with a 2-minute exposure (Bruker). The following week these animals were then treated with 1 mg/kg LPS for 16 hours. The mice were then intraperitoneally injected with 100 µL of 200 µM FFA-SS-luc (0.014 mg/mouse) bound to 0.1% (w/v) BSA in PBS immediately prior to imaging. Luminescent images were acquired with a 2-minute exposure (Bruker).

**Analysis of CPT1A in vivo**. CPT1A is an enzyme located on the outer mitochondrial membrane and responsible for catalyzing the first step during mitochondrial fatty-acid oxidation. C57BL/6 J mice were left untreated or infected with 100 µl of $1 \times 10^8$ CFU *S. typhimurium* (SL1344- JH3009) by oral gavage or infected with *S. typhimurium* by oral gavage and treated with etomoxir interperitoneally, etomoxir was repeatedly administered every 24 hrs. After 72 hours of *S. typhimurium* infection mice were killed by exposure to $CO_2$. The BM was extracted for flow cytometry-based lineage analysis. C57BL/6 J mice were also pre-treated with 50 mg/kg etomoxir interperitoneally for 1 hour followed by 1 mg/kg LPS interperitoneally after 16 hours the mice were killed, and the BM was extracted for flow cytometry-based lineage analysis of the LSK and HSC populations.

**FFA in serum**. C57Bl/6 J mice were left untreated or infected with 100 µl of $1 \times 10^8$ CFU *S. typhimurium* (SL1344- JH3009) by oral gavage for 72 hours. C57Bl/6 J mice were treated with LPS or control PBS for 16 hours. IL-6 inhibitor (clone MP5-20F3) was used in combination with LPS. The mice were anesthetized and 600 µl blood was taken by cardiac puncture. The blood was spun at $1600 \times g$ for 10 mins to remove cells, the supernatant was then spun again at $16,000 \times g$ for 5 mins to collect serum. The serum was analyzed for free fatty acids using Plasma Fatty Acid and Glycerol Detection Kits (Zenbio, US) as per the manufacturer's instructions.

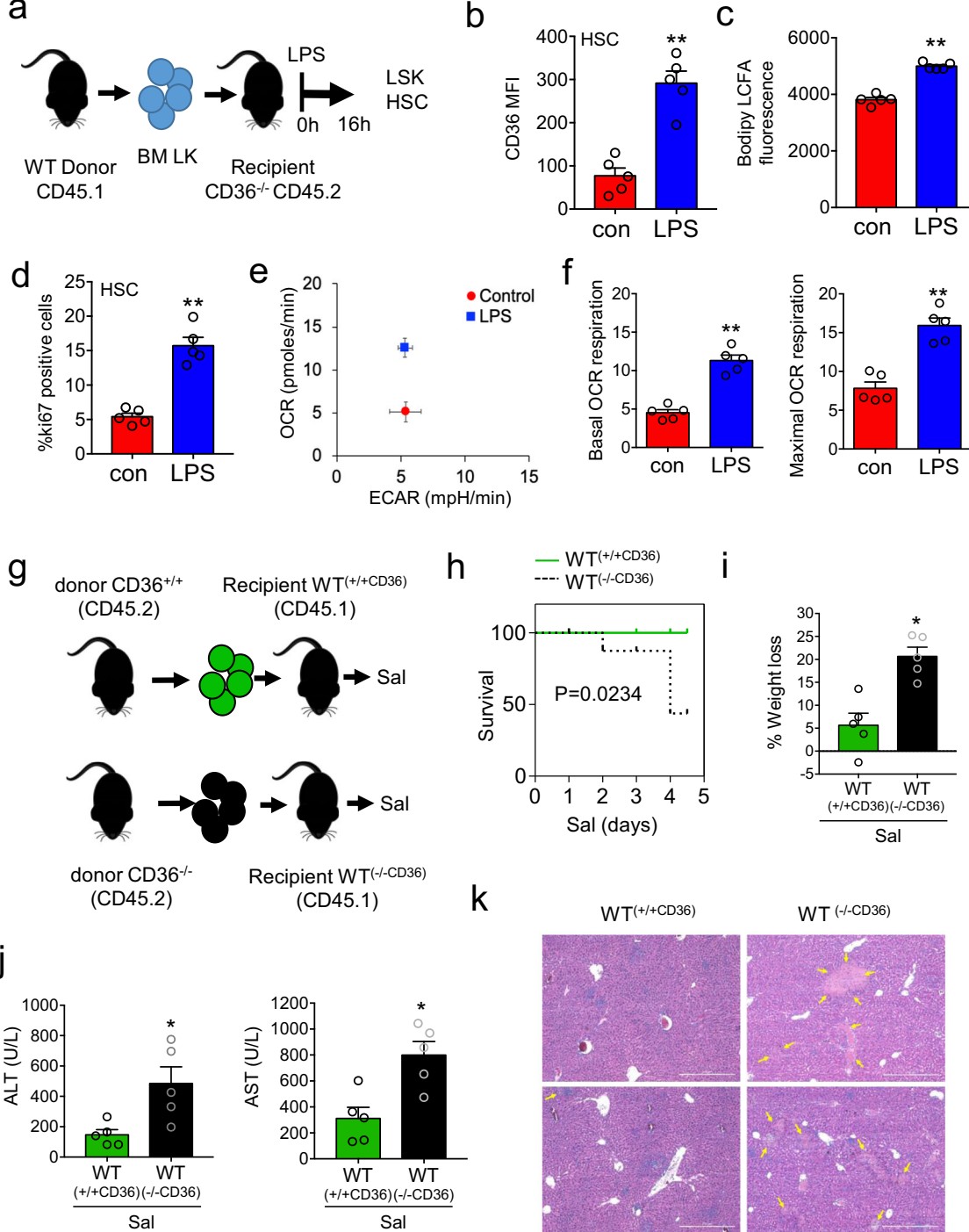

. C57Bl/6 J or CD36$^{+/+}$ or CD36$^{-/-}$ or transplant mice were interperitoneally injected with 1 mg/kg LPS after 16 hours the mice were sacrificed. The BM extracted, and the mouse Lineage negative cells were enriched using a direct cell lineage depletion kit and CD117+ enrichment kit (Miltenyi Biotec, Germany). In all, $5 \times 10^4$ LK cells were incubated with a long-chain fatty-acid 4,4-Difluoro-5,7-Dimethyl-4-Bora-3a,4a-Diaza-s-Indacene-3-Dodecanoic Acid (BODIPY™ FL C$_{12}$) (1 μM, Invitrogen) at room temperature for 20 min, washed twice in 1× PBS and centrifuged at $400 \times g$ for 5 minutes before resuspending in PBS in a glass-bottom 96-well plate and read on a plate reader at 558/568 on a BMG Labtech microplate reader.

**Flow cytometry and cell sorting**. Antibody cocktails were prepared in MACs buffer and incubated with BM cells for at least 30 min at 4 °C. Experiments using 4-difluoro-1,3,5,7,8-pentamethyl-4-bora-3a,4a-diaza-s-indacene (BODIPY 493/503), the cells were incubated with BODIPY 493/503 (1 μM, Invitrogen) at room temperature for 20 min, washed twice in 1× PBS and centrifuged at $400 \times g$ for

5 minutes before staining with antibody cocktail. For experiments using Ki67 the BM cells were incubated with an antibody cocktail prior to fixing and permeabilization using the FIX & PERM™ Cell Permeabilization Kit (ThermoFisher, Waltham, MA, USA) as per the manufacturer's instructions. The cells were then stained with Anti-Ki-67-FITC, human and mouse (Miltenyi Biotec, Germany) for 20 min centrifuged at $400 \times g$ for 5 mins before resuspending in PBS. For flow cytometric cell sorting of BM cell populations pellet was resuspended in antibody mix and cells were sorted directly into lysis buffer. Flow cytometry was carried out using FACSCanto II flow cytometer (BD Bioscience) and cell sorting was performed on a BD FACSMelody (BD Bioscience). Data were collected using BD FACS Diva software 8.0.1. Data were analyzed using FlowJo 10.7.0 (TreeStar, Ashland, OR, USA). See figures for specific gating strategies.

Flow cytometry antibodies were purchased from Biolegend, Miltenyi Biotec and ThermoFisher. Lineage cocktail Pacific Blue anti-mouse Biolegend Catalog number:133310 Lot number: B289722 (dilution 1:50). Components include anti-mouse CD3, clone 17A2 (dilution 1:100); anti-mouse Ly-6G/Ly-6C, clone RB6-8C5

**Fig. 5 FFA uptake through CD36 is an essential component of HSC expansion in response to infection. a** Schematic diagram of experimental design. $CD36^{+/+}$ CD45.1 lineage negative, CD117-positive cells were isolated and transplanted into $CD36^{-/-}$ CD45.2 animals. Post engraftment mice were treated with 1 mg/kg LPS for 16 hours and the bone marrow cells were analyzed by flow cytometry. **b** Flow cytometry analysis of CD36 expression (CD36 mean fluorescence intensity (MFI)) in the HSC from the transplant mice following 1 mg/kg LPS (16 hours) treatment. $n = 5$ mice in each group. **c** The LK cells were isolated and long-chain fatty-acid (LCFA) uptake was measured using the QBT assay. $n = 5$ in each group. **d** Percentage of cycling HSCs as measured by Ki67-positive cells from transplant mice after 16 hours of 1 mg/kg LPS treatment. $n = 5$ mice in each group. **e** The LSK population was isolated by FACS, oxygen consumption rate (OCR) was measured by the extracellular flux assay. Basal extracellular acidification rate (ECAR) compared with basal OCR levels provides a snapshot of the bioenergetic profile of LSK before and after treatment with LPS (16 hours). Basal OCR normalized to rotenone. $n = 5$ mice in each group. **f** Basal (normalized to rotenone) and maximal mitochondrial respiration in LSK cells from control vs. LPS 16 hours treated transplanted mice. $n = 5$ mice in each group. **g** $CD36^{+/+}$ CD45.2 or $CD36^{-/-}$ CD45.2 lineage negative, CD117-positive cells were isolated and transplanted into WT CD45.1 mice these were termed $WT^{(+/+CD36)}$ or $WT^{(-/-CD36)}$. Post engraftment mice were treated with *S. typhimurium* for 96 hours. **h** Kaplan–Meier survival curve $n > 5$ mice in each group. **i** Weight loss was analyzed. $n = 5$ in each group. **j** Levels of circulating alanine aminotransferase (ALT) and aspartate aminotransferase (AST) in the serum following 96 hours of *S. typhimurium* treatment. $n > 5$ in each group. **k** Livers were isolated and sectioned and stained with hematoxylin and eosin. (Magnification: H, ×63). $n = 5$ mice in each group. Data shown are means ± SD. The Mann–Whitney U test (two-tailed) was used to compare between treatment groups *$p < 0.05$ **$p < 0.01$. Source data are provided as a Source Data file.

(dilution 1:100); anti-mouse CD11b, clone M1/70 (dilution 1:100); anti-mouse CD45R/B220, clone RA3-6B2 (dilution 1:50); anti-mouse TER-119/Erythroid cells, clone Ter-119 (dilution 1:50). CD117 Pe-Vio 770 anti-mouse Miltenyi Biotec Catalog number:130-111- 695 Lot number: 5200705289 Clone: REA791 (dilution 1:50). Sca1-APC mouse Miltenyi Biotec Catalog number:130-123-848 Lot: 5191212189 Clone: REA422 (dilution 1:50). CD150 (SLAM) Brilliant Violet 510 anti-mouse Biolegend Catalog number:115929 Lot number: B264811 Clone: TC15-12F12.2 (dilution 1:50). CD48 APC-Cyanine 7 Anti-mouse Biolegend Catalog number: 103432 Lot number: B271042 Clone: HM48-1 (dilution 1:50). CD34 PerCP/Cyanine 5.5 Anti-mouse Biolegend Catalog number: 128608 Lot number: B271491 Clone: HM34 (dilution 1:50). CD36 VioBright 515 mouse Miltenyi Biotec Catalog number: 130-122-094 Lot number: 520020623 Clone: REA1184 (dilution 1:50). Anti-Ki67-FITC human and mouse Miltenyi Biotec Catalog number: 130-100-339 Clone: REA183 Lot number: 5171102132 (dilution 1:100). CD45.1 Antibody, anti-mouse, APC Miltenyi Biotec Catalog number: 130-102-470 Clone: A20 (dilution 1:200). CD45.1 Antibody, anti-mouse, PE Miltenyi Biotec Catalog number: 130-102-499 Clone: A20 (dilution 1:200). CD45.2 Antibody, anti-mouse, FITC Miltenyi Biotec Catalog number: 130-102-997 Clone: 104-2 (dilution 1:200). Anti-FABP3 clone and catalog number PA5-13461, ThermoFisher (dilution 1:50).

**Real-time PCR for lipid uptake genes.** ReliaPrep RNA cell miniprep system (Promega, Southampton, UK) was used to extract whole cell RNA. Nugen PicoSL WTA (Redwood City, USA) was used to generate the first stand, followed by double-strand cDNA, followed by cDNA amplification. qRT-PCR assay was performed with the SYBR-green technology (PCR biosystems, UK). PCRs were amplified for 45 cycles (95 °C/15 seconds, 60 °C/10 seconds, 72 °C/10 seconds), after pre-amplification (95 °C/60 seconds), on a Roche 384-well/96-well Light-Cycler480. Messenger RNA (mRNA) expression was normalized against glyceraldehyde 3-phosphate dehydrogenase using the comparative cycle threshold method. KiCqStart® SybrGreen Primers were purchased from Sigma-Aldrich and sequences are shown in Supplementary Table 1.

**Confocal immunofluorescent microscopy.** BM from *S. typhimurium* (72 hours) infected C57BL/6 J, $CD36^{+/+}$, and $CD36^{-/-}$ animals were lineage depleted and CD117 enriched (LK). These LK cells were then stained with 5 μM Hoechst 33342 (ThermoFisher, Waltham, MA, USA) and a fluorescent neutral lipid dye 4-difluoro-1,3,5,7,8-pentamethyl-4-bora-3a,4a-diaza-s-indacene (BODIPY 493/503) (1 μM) (ThermoFisher, Waltham, MA, USA) at room temperature for 30 minutes. The cells were washed twice in 1× PBS by centrifugation at $400 \times g$ for 5 minutes and stained with the cell surface antibody Sca1-APC (Miltenyi Biotec, Bergisch Gladbach, Germany) for 20 minutes in the dark. The LK cells positive for Sca1 marker were then termed LSKs. The cell suspension was washed again with 1× PBS and resuspended in 200 μL of FluoroBrite DMEM medium supplemented with 10% fetal calf serum (FCS) and plated in a black-walled imaging plate. Confocal images were acquired on a Zeiss LSM 800 Axio Observer.Z1 using a ×63 water objective and data were collected on Zenbio. Fiji imageJ 2.0.0 software was used for image processing.

**Seahorse.** XFp flux cartridges were hydrated in XF Calibrant overnight at 37 °C. LK were isolated from the BM 5 mice either treated with LPS or PBS or *S. typhimurium* by magnetic bead separation (Miltenyi Biotec, Germany) into conventional Seahorse base media supplemented with pyruvate (1 mM), L-glutamine (2 mM), Glucose (10 mM). In all, 100,000 cells were plated onto one well of a Seahorse XFp culture plate coated with poly-D lysine and centrifuged briefly to achieve a uniform monolayer of cells. Cells were equilibrated in a humidified non-$CO_2$ incubator until the start of the assay. Flux cartridges were loaded with Oligomycin (2 μM), carbonyl cyanide-4-(trifluoromethoxy) phenylhydrazone (1 μM), and Rotenone (0.5 μM) according to manufacturer's instructions. Oxygen

consumption rate and basal extracellular acidification rate values were obtained using the XFp Mito Stress Kit.

For fatty-acid fuel dependency, flux cartridges were loaded according to the manufacturer's instructions. Fatty-acid dependency, capacity, and flexibility values were obtained using the XFp Mito Fuel Flex Test Kit. Metabolic parameters were derived from calculations based on the manufacturer's instructions. All results were normalized to input cell numbers. Owing to the feasibility of cell number isolation, experiments are represented as one replicate per cell type, per condition. Data were analyzed using Wave 2.6.1.

***S. typhimurium* and LPS-elicited stressed hematopoiesis.** Glycerol stock of *S. typhimurium* (SL1344- JH3009) was a kind gift from Dr. Isabelle Hautefort (Quadram Institute Bioscience, Norwich). The stock was plated on Luria Broth agar plates and the colonies were inoculated and grown overnight into 5 ml of Luria Broth with 0.3 M NaCl (LBS). The overnight culture was then diluted 1:100 in LBS and grown until the culture optical density ($\Delta OD_{600nm}$) of 1.2–1.4 (late exponential phase). This is the time point where SPI1 invasion genes are turned on in *S. typhimurium*. The bacterial culture was then centrifuged at $3000 \times g$ for 7 minutes before washing bacterial cells twice in 25 ml of sterile DPBS at room temperature. Finally, resuspend the bacterial cells in sterile DPBS at a concentration of $1–5 \times 10^8$ CFU per 100 μl of DPBS (knowing that $DOD_{600nm}$ 1.26 corresponds to $7.53 \times 10^8$ CFU/ml).

C57Bl/6 J or $CD36^{+/+}$ or $CD36^{-/-}$ mice were treated with streptomycin (20 mg/ml) 24 hr prior to *S. typhimurium* infection. Mice were then left untreated or infected with 100 μl of $1 \times 10^8$ CFU *S. typhimurium* (SL1344-JH3009) by oral gavage for 72 hours. C57Bl/6 J, CD36 +/+, CD36−/− or transplanted mice were treated with 1 mg/kg LPS intraperitoneally or control PBS for 2 or 16 hours. C57Bl/6 J mice were also subjected to pre-treatment of 40 mg/kg SSO intraperitoneally or 10 mg/kg Etomoxir intraperitoneally for 1 hour followed by 1 mg/kg LPS. The mice were sacrificed by exposure to $CO_2$, and the BM was analyzed by flow cytometry, cell sorting, and seahorse metabolic analysis.

**Liver histology.** Liver tissues were harvested and immediately fixed in 10% neutral formalin and embedded in paraffin blocks 24 hours later. Tissue blocks were sectioned, dewaxed, and hydrated prior to being stained with Hematoxylin & Eosin (H&E) for histopathological analysis.

**Serum transaminases.** The levels of circulating ALT and AST were measured in serum samples in a Randox RX Daytona analyser.

**Quantification and statistical analysis.** For statistical comparison of more than two groups were compared, Kruskal–Wallis test followed by Dunn's multiple comparisons or two-way analysis of variance was performed using Prism version 7.00 for Windows (GraphPad, La Jolla, CA, USA). Owing to variability in the data, statistical comparison of in vivo work was performed without assumption of normal distribution using Mann–Whitney test. Differences among group means were considered significant when the probability value, $p$, was <0.05*, 0.01**, 0.001***. Sample size ($n$) represents a number of biological replicates. No statistical methods were used to predetermine sample size.

**Reporting summary.** Further information on research design is available in the Nature Research Reporting Summary linked to this article.

## Data availability
The data generated in this study are provided in the Supplementary Information. Source data are provided with this paper.

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

## Acknowledgements

S.A.R. received funding from the Norwich Research Park (NRP), The Rosetrees Trust, and The Big C. The work was supported by the MRC project grant S.A.R. (MR/T02934X/1). C.H. was funded by the Wellcome Trust clinical fellowship program. We also acknowledge Earlham Institute using support from the UK Research and Innovation (UKRI) Biotechnology and Biological Sciences Research Council (BBSRC) under grants National Capability in Genomics and Single Cell "BBS/E/T/000PR9816. The authors also thank Dr. Allyson Tyler and Dr. Karen Ashurst from the Laboratory Medicine Department at the Norfolk and Norwich University Hospital for technical assistance. We thank the team at the Disease Modeling Unit of the University of East Anglia for assistance with the in vivo studies. pCDH-LucferaseT2A-mCherry was kindly gifted by professor Irmela Jeremias, MD, from Helmholtz Zentrum München, Munich, Germany.

## Author contributions

J.J.M., K.M.B, N.B., C.H., and S.A.R. designed the research, analyzed the data, and wrote the paper; J.J.M., C.H., J.A.M., and A.J., conducted experimental design and executed most of the experiments; J.J.M., J.A.M., N.B., M.M.-G., S.A.R., and C.H. carried out in vivo work; I.M., F.D.-P., and K.M.B. provided essential reagents and knowledge.

## Competing interests

The authors declare no competing interests.
