## [Peer Review File · Nature Communications]

Free fatty acid transport via CD36 drives β -oxidation mediated hematopoietic stem cell response to infectionReviewers' comments:

Reviewer #1 (Remarks to the Author):

Upon acute infections quiescent HSCs leave their quiescent stage, enter the cell cycle and start to proliferate. The mechanisms involved remain largely unknown, however it has been shown already that the transition from dormant to active HSCs is accompanied by a metabolic switch, with quiescent HSCs being metabolically inactive (Liang et al, and Hinge et al, Cell Stem Cell, 2020). These papers should be cited in this manuscript. Here, they focus on the free fatty acid uptake by HSCs during acute infection and suggest a role for CD36 in this process. The data shown is very interesting, however differences are not very pronounced and there are some concerns regarding technical aspects of the experiments. In addition, the authors need to make sure the figure legends have all the necessary information so the reader does not have to puzzle together information from the legend and the main text. Some details on how the experiments are performed are also lacking. And make sure to use the same abbreviations in the text and the figures (for example eto vs ETX).

Major comments:

- gating within the stem cell compartment upon acute inflammation: it has been described by many groups investigating stem cells under inflammation that marker expression is changing in these conditions, which is also shown by the authors in the FACS plots in figure 1e. This makes it very difficult to gate for the different populations and be sure of the cell types you think you are looking at. Additional markers such as EPCR and CD34 should be used to gate for the quiescent HSC compartment and marker changes should be considered in conclusions on LSK and MPP populations. In addition, FACS data should be shown for many of the analysis. Only then will it be possible to judge the data correctly.
- Why were the time-points 16h post LPS and 72h post Sal chosen? Were timecourse kinetic experiments performed? And what is the rationale for comparing the different timepoints for the 2 conditions? What is the evidence that we are looking at the same phase of the inflammatory response? The data in figure 4 for example would suggest that the LPS response might even be faster than 16h with the 2h timepoint having most changes in expression of transporter proteins
- In many (if not all) of the experiments with transplanted animals treatments are already started 4 weeks post transplantation, when the hematopoietic system is still in its initial reconstitution phase, the reconstituting system is probably still in a stress state and HSCs do not show stable engraftment yet. Thus, these conditions will highly likely influence the analysis. Since the goal of the manuscript is to analyse the stem cell compartment, one should wait at least 12-16 weeks post transplant for stable engraftment with the start of the treatments.
- In many experiments the differences are not big and variation is quite high. It would be more informative to show individual data points in all the graphs rather than bar charts. Also the question is how biologically relevant the small differences observed are.

- Revise statistical testing. One-Way-ANOVA is not appropriate when there are 2 variables (e.g. fig. 2 e,f). Results of One-Way-ANOVA should be listed in figure legend, and it should be clearly defined what the star to indicate statistical significance is referring to when there are more than two groups in a panel.
- Fig. 3A: Why is ETX given together with Sal, but in the LPS model it is given 1 h before treatment? These results are most likely not comparable, as the metabolic state at time of stimulation is indeed critical. In the LPS model the cells are already preexposed to ETX, whereas in the Sal model they are not.
- In Figure 3D CPT1A pops up out of the blue, without any explanation or background information. And what is the p-value and statistical test here? It looks like a ~60fold increase from CTR to LPS, with relatively small error bars, so a $0.01 > p < 0.5$ seems improbable. Also, the axis should be cut less close to the mean of the LPS bar (e.g. a cut at ~2.5 seems more appropriate). Why is the data for Sal not shown?
- The data in Fig. 3F need to be further explained: is this a mere transplant effect (e.g. Ki67 levels are very different in the CTR group compared to values shown in panel C and D)? The extent of β -oxidation at baseline and post inflammatory stimulation should be shown in CTR and KD cell recipients. What is the efficiency of KD? What is the role of lenti-viral transduction of the cells. Were CTR cells also treated with a lenti virus?
- Fig. 4: It is surprising that while multiple FFA transporters seem to be induced upon Sal and LPS treatment (Fig 4A), inhibition of a single transporter (CD36) is reducing Ki67 levels back to CTR levels. Does this mean all other transporters are dysfunctional? Or is SSO so unspecific? SSO specificity should be demonstrated (e.g. showing no changes in FFA uptake between CD36 sufficient and deficient cells)
- Where are the control animals in Figure 4 G and H? only KO animal data are shown. And why is ki67 even lower in LPS treated animals? I.e. this reduction seems even more pronounced than the induction in WT cells they have shown previously. Also, this does not recapitulate the SSO data in panel E
- Fig 4 M: when and how long were these cells/mice treated with LPS? Figure legend needs to give ALL necessary information! Is engraftment the same? Is the composition of cells that are transplanted the same?
- The chimeras shown in Figure 5a-f are interesting, however only show that CD36 expression in the stroma is not essential, not that CD36 lacking in HSCs is sufficient. For this they would need to repeat the experiments by transplanting CD36KO cells into wt animals, which is partially shown in the rest of the figure. Question here is when the analysis were performed, whether there was a difference in reconstitution of wt vs CD36KO cells and how the observed liver phenotype is linked to CD36 expression in HSCs?

Specific comments:

- Figure S1 A/B, there really is only a marginal increase in FFA in the serum. Is this biologically relevant?
- There are two D-panels in figure 1
- Figure 1 i: indicate what you are staining directly in the panel
- Line 115: infected "with" Sal (with is missing)

- Figure 2b: show actual ECAR seahorse plot. What are you looking at? Baseline glycolysis, maximal glycolysis? Where are also the OCR values from? From Panel a, I can't spot a measurement where OCR ($\mu\text{mole}/\text{min}$) is 5 in CTR and 10 in Sal treated samples. Same is true for the LPS.
- Figure 2c: are the stats missing or is there no statistical significance? Is this small increase biologically relevant?
- Revise depiction of cells isolated from animals. Currently these cartoons are reminiscent of sperm, rather than HSPCs
- Line 128: figure number is missing
- Why are there not data on Sal treatment in the second part of Figure 2?
- A quantification of the data in Figure 4I would be informative

Reviewer #2 (Remarks to the Author):

This study reports that induction of Cd36 and uptake of free fatty acids are required for the transition to fatty acid oxidation necessary for expansion of hematopoietic stem cells (HSC) during infection.

The authors and others have previously reported on the role of FA oxidation in expansion and differentiation of HSCs including in response to infection. The new finding here is that the mechanism for the oxidative transition is uptake of long chain fatty acids mediated by induction of CD36. This finding is in line with CD36's influence on the cellular shift to FA oxidation documented in many cell types. The authors validate their finding using several approaches including a nice mouse model that allows visualization of the uptake.

The main finding is interesting, however, considering what is already reported in the literature, additional work would be required to increase the novelty of the study.

Comments:

1-What cytokine/growth factor might be driving CD36 induction on HSC at the onset of infection? Do FA derived from BM adipose tissue play a role?

2-CD36 has functions other than FA uptake such as recognition and internalization of pathogens. Its deletion has been previously shown to decrease survival during infection (Stuart LM JCB 2005) which reflected its role in phagocytosis. This complicates interpretation of the survival studies in transplanted mice in terms of the role of HSC FA oxidation.

3-If FA oxidation is making HSC more competent against infection, it seems that this could be duplicated with short chain fatty acids which are oxidized and do not require CD36.

4- The discussion is very short and does not provide much insight into the relevance of the findings and their novel contribution to our current knowledge.

Reviewer #3 (Remarks to the Author):

The manuscript by Mistry et al focuses on the role of FFA in HSC functions following infection. The authors suggest that infection facilitates FFA uptake via CD36 and a metabolic switch to beta-oxidation. While, the subject matter is indeed of interest, there are many gaps in this paper that need to be addressed. They are listed below.

- 1) The authors claim to have developed a novel assay for FFA uptake (Figure 1C-D). It would be important to describe the principle of this assay in greater detail.
- 2) There are two Figures 1D. Please correct.
- 3) The FACS plots shown in Figure 1E are difficult to interpret. There seems to be a technical issue with this experiment. C-Kit staining does not seem to work properly and the separation of the LSK population is not convincing (i.e. LSK cells are hardly visible). Furthermore, the CD150/CD48 staining shown does not allow for a confident separation of HSCs. In summary, the LSK and HSC staining needs to be optimised.
- 4) The data shown in Figure 2A and D display OCR of LSK cells. However, no statistical analyses (P values) are shown. Furthermore, statistical analyses are not offered for results shown in Figure 2C.
- 5) In some cases, etomoxir is referred to as Eto, in other cases as ETX. This needs to be corrected.
- 6) Given a possible technical issue with LSK/HSC staining, it would be important that the authors display the LSK/HSC FACS plots and the Ki67 staining.
- 7) When referring to the data shown in Figure 3B-C, they state that HSCs are undergoing expansion. It would be important to also display the total numbers of LSKs and HSCs (e.g. per leg or per bone) to document the actual change in cell numbers.
- 8) The authors state: "Mice received adoptive transfer of WT (LK con KD) or CPT1A KD (LKCPT1A KD)..." – the exact details of this experiment need to be provided. The knockdown efficiency of CPT1A must to be shown by WB or qPCR.
- 9) In Figure 4, CD36 KO mice are used. No details about any possible phenotypes are provided. It would be highly appropriate to treat CD36 WT and KO mice with LPS and control and perform detailed analyses

of haematopoiesis at different timepoints and bone marrow HSCs and progenitor cell compartments. It would also be appropriate to sort HSCs from these mice and transplant them to test their multilineage reconstitution potentials. It will also be important to analyse the cell cycle profile in HSCs from untreated and treated CD36 WT and KO mice. Multiple functional assays are missing and therefore this study is premature and underdeveloped.

10) Again, Figure 4J shows Seahorse data but no statistical analyses are offered. Are there no significant changes between the groups shown in this figure?

11) The experiments shown in Figure 5A are not introduced in sufficient detail to allow the full appreciation of the results presented here.

12) The authors state: “These findings provide mechanistic understanding of the interplay between HSC and the bone marrow microenvironment” – where are the data supporting this interplay?

Response to reviewers

Upon acute infections quiescent HSCs leave their quiescent stage, enter the cell cycle and start to proliferate. The mechanisms involved remain largely unknown, however it has been shown already that the transition from dormant to active HSCs is accompanied by a metabolic switch, with quiescent HSCs being metabolically inactive (Liang et al, and Hinge et al, Cell Stem Cell, 2020). These papers should be cited in this manuscript. Here, they focus on the free fatty acid uptake by HSCs during acute infection and suggest a role for CD36 in this process. The data shown is very interesting, however differences are not very pronounced and there are some concerns regarding technical aspects of the experiments. In addition, the authors need to make sure the figure legends have all the necessary information so the reader does not have to puzzle together information from the legend and the main text. Some details on how the experiments are performed are also lacking. And make sure to use the same abbreviations in the text and the figures (for example eto vs ETX).

Thank you for your comments. We have now addressed the technical concerns you highlighted below in your review. We have also expanded the figure legends and methods to include more detail regarding how the experiments were performed.

Major comments:

- gating within the stem cell compartment upon acute inflammation: it has been described by many groups investigating stem cells under inflammation that marker expression is changing in these conditions, which is also shown by the authors in the FACS plots in figure 1e. This makes it very difficult to gate for the different populations and be sure of the cell types you think you are looking at. Additional markers such as EPCR and CD34 should be used to gate for the quiescent HSC compartment and marker changes should be considered in conclusions on LSK and MPP populations. In addition, FACS data should be shown for many of the analysis. Only then will it be possible to judge the data correctly.

Response: we have now added representative plots for our FACs data for control and salmonella treated HSPC in figure 1F, and included new data showing CD34 in our analysis to help gate quiescent HSC compartment. In addition, FACs data is now provided for experiments in Figure 1F, 3H (ki67), Supplementary figures, 4 (Cd36 expression in LSK and HSC), 6 (CD45.1 chimerism in CD45.2 transplanted animals), 8 (myeloid/lymphoid ratios in transplanted animals) and 9 (HSPC engraftment in transplanted animals).

- Why were the time-points 16h post LPS and 72h post Sal chosen? Were timecourse kinetic experiments performed? And what is the rational for comparing the different timepoints for the 2 conditions? What is the evidence that we are looking at the same phase of the inflammatory response? The data in figure 4 for example would suggest that the LPS response might even be faster than 16h with the 2h timepoint having most changes in expression of transporter proteins

Response: to help us determine the best time to study the FFA uptake response in vivo, we used IL-6 expression as a surrogate marker (data now shown in supplementary figure 1A). This rationale is now explained in data in supplementary figure 1. We recognise that this is not a perfect surrogate, but IL-6 has been linked to lipolysis in various studies (<https://doi.org/10.1210/jc.2002-021687> and DOI: 10.1007/s12020-008-9085-7) and therefore gives us an indicative time point for which to assess activation of the immune response.

- In many (if not all) of the experiments with transplanted animals treatments are already started 4 weeks post transplantation, when the hematopoietic system is still in its initial reconstitution phase, the reconstituting system is probably still in a stress state and HSCs do not show stable engraftment yet. Thus, these conditions will highly likely influence the analysis. Since the goal of the manuscript is to analyse the stem cell compartment, one should wait at least 12-16 weeks post transplant for stable engraftment with the start of the treatments.

Response: We apologise but this was a mistake in the text. We have now corrected this mistake and provided data to show chimerism in CD45.2 animals transplanted with CD45.1 LK cells (supplementary figure 6). All experiments on transplanted animals were performed at least 12 weeks post injection of cells.

- In many experiments the differences are not big and variation is quite high. It would be more informative to show individual data points in all the graphs rather than bar charts. Also the question is how biologically relevant the small differences observed are.

Response: Working with in vivo systems can cause variations in the data. We have tried to limit the variations by using same sex and similar aged mice for individual experiments. We have now put the individual dots on each bar graph to show variation between mice in experimental groups.

- Revise statistical testing. One-Way-ANOVA is not appropriate when there are 2 variables (e.g. fig. 2 e,f). Results of One-Way-ANOVA should be listed in figure legend, and it should be clearly defined what the star to indicate statistical significance is referring to when there are more than two groups in a panel.

Response: this has now been corrected

- Fig. 3A: Why is ETX given together with Sal, but in the LPS model it is given 1 h before treatment? These results are most likely not comparable, as the metabolic state at time of stimulation is indeed critical. In the LPS model the cells are already preexposed to ETX, whereas in the Sal model they are not.

Response: We apologise but this was a mistake in figure 3A ETO was given before Salmonella infection. This has now been corrected.

- In Figure 3D CPT1A pops up out of the blue, without any explanation or background information. And what is the p-value and statistical test here? It looks like a ~60fold increase from CTR to LPS, with relatively small error bars, so a $0.01 > p < 0.5$ seems improbable. Also, the axis should be cut less close to the mean of the LPS bar (e.g. a cut at ~2.5 seems more appropriate). Why is the data for Sal not shown?

Response: CPT1A has now been introduced in the text (page 6 para 1). We have now performed the experiment on Salmonella treated animals and the data included in Figure 3F. The p values have been included in the legend. The difference in expression of CPT1a between Salmo and LPS is interesting and probably due to differences in models to induce HSC cycling.

- The data in Fig. 3F need to be further explained: is this a mere transplant effect (e.g. Ki67 levels are very different in the CTR group compared to values shown in panel C and D)? The extent of b-oxidation at baseline and post inflammatory stimulation should be shown in CTR and KD cell recipients. What is the efficiency of KD? What is the role of lenti-viral transduction of the cells. Were CTR cells also treated with a lenti virus?

Response: The flow figures are now shown for this figure to show gating. We do see lower levels of ki67% cells. This may be due to analysing the transplanted cells and not all HSC, as we did in figure c and d. The knockdown efficiency is now included in supplementary figure 2C along with engraftment data. The legend now states that both control KD and CPT1A KD were treated with virus.

- Fig. 4: It is surprising that while multiple FFA transporters seem to be induced upon Sal and LPS treatment (Fig 4A), inhibition of a single transporter (CD36) is reducing Ki67 levels back to CTR levels. Does this mean all other transporters are dysfunctional? Or is SSO so unspecific? SSO specificity should be demonstrated (e.g. showing no changes in FFA uptake between CD36 sufficient and deficient cells)

Response: This is a good question and something we hadn't considered. We have now performed an SSO only experiment to determine if the inhibitor has any effect without LPS. Supplementary figure 4B shows that SSO had no effect on Ki67 cells under control conditions.

- Where are the control animals in Figure 4 G and H? only KO animal data are shown. And why is ki67 even lower in LPS treated animals? I.e. this reduction seems even more pronounced than the induction in WT cells they have shown previously. Also, this does not recapitulate the SSO data in panel E

Response: The control animal data is now included in this figure. We agree with the reviewer that the Ki67 data was even more pronounced in the CD36 knockout animals. Further experiments showed that this difference was caused by problems with compensation controls. We therefore performed these experiments again the data for which is shown in figure 4G and H.

- Fig 4 M: when and how long were these cells/mice treated with LPS? Figure legend needs to give ALL necessary information! Is engraftment the same? Is the composition of cells that are transplanted the same?

Response: More detail is given in the legend to address this comment. Additional data is shown in figure supplementary figure 5 D and E.

- The chimeras shown in Figure 5a-f are interesting, however only show that CD36 expression in the stroma is not essential, not that CD36 lacking in HSCs is sufficient. For this they would need to repeat the experiments by transplanting CD36KO cells into wt animals, which is partially shown in the rest

of the figure. Question here is when the analysis were performed, whether there was a difference in reconstitution of wt vs CD36KO cells and how the observed liver phenotype is linked to CD36 expression in HSCs?

Response: We repeated the experiment as the reviewer suggested and examined reconstitution of wt vs CD36KO cells. Supplementary figure 7E shows that wt and CD36KO cells have similar engraftment in to CD45.1 animals.

Specific comments:

- Figure S1 A/B, there really is only a marginal increase in FFA in the serum. Is this biologically relevant?

Response: This is difficult to answer when we are only looking at one time point. Especially since we have shown that FFA are being taken up by cells of the haematopoietic system. Therefore, any major observational increase may be offset by uptake.

- There are two D-panels in figure 1

Response: This has been amended

- Figure 1 i: indicate what you are staining directly in the panel

Response: the legend and panel have been amended.

- Line 115: infected "with" Sal (with is missing)

Response: corrected

- Figure 2b: show actual ECAR seahorse plot. What are you looking at? Baseline glycolysis, maximal glycolysis? Where are also the OCR values from? From Panel a, I can't spot a measurement where OCR (pmole/min) is 5 in CTR and 10 in Sal treated samples. Same is true for the LPS.

Response: Basal OCR was measured before Oligomycin injection and maximum OCR after FCCP injection, with non-mitochondrial respiration subtracted. This is why the numbers look different between these panels.

- Figure 2c: are the stats missing or is there no statistical significance? Is this small increase biologically relevant?

Response: Stats now included.

- Revise depiction of cells isolated from animals. Currently these cartoons are reminiscent of sperm, rather than HSPCs

Response: Revised

- Line 128: figure number is missing

Response: corrected

- Why are there not data on Sal treatment in the second part of Figure 2?

Response: we just performed this analysis on LPS treated animals

- A quantification of the data in Figure 4I would be informative

Response: The quantification is the data in figure 4G – which is by flow. This is more sensitive than using microscopy images and performing data analysis using imageJ

Reviewer #2 (Remarks to the Author):

This study reports that induction of Cd36 and uptake of free fatty acids are required for the transition to fatty acid oxidation necessary for expansion of hematopoietic stem cells (HSC) during infection.

The authors and others have previously reported on the role of FA oxidation in expansion and differentiation of HSCs including in response to infection. The new finding here is that the mechanism for the oxidative transition is uptake of long chain fatty acids mediated by induction of CD36. This finding is in line with CD36's influence on the cellular shift to FA oxidation documented in many cell types. The authors validate their finding using several approaches including a nice mouse model that allows visualization of the uptake.

The main finding is interesting, however, considering what is already reported in the literature, additional work would be required to increase the novelty of the study.

Thank you for taking time to review our manuscript. Below is our responses to your comments

Comments:

1-What cytokine/growth factor might be driving CD36 induction on HSC at the onset of infection? Do FA derived from BM adipose tissue play a role?

Response: This is an interesting point and something we intend to do in the future. We have shown that IL-6 levels mimic the timepoints taken for LPS and Salmonella. Moreover, Il-6 has been shown to induce lipolysis in various studies (<https://doi.org/10.1210/jc.2002-021687> and DOI: 10.1007/s12020-008-9085-7). We have added discussion on this important point page 11 para 2.

2-CD36 has functions other than FA uptake such as recognition and internalization of pathogens. Its deletion has been previously shown to decrease survival during infection (Stuart LM JCB 2005) which reflected its role in phagocytosis. This complicates interpretation of the survival studies in transplanted mice in terms of the role of HSC FA oxidation.

Response: We agree with the reviewer. This is why we used 3 different systems to show CD36 is important in regulating FFA uptake in HSC. First, we used the CD36 inhibitor SSO, second we used CD36 ko mice and finally we used transplanted CD36 HSPC into wild type animals. We have added discussion on this point page 11 para 1.

3-If FA oxidation is making HSC more competent against infection, it seems that this could be duplicated with short chain fatty acids which are oxidized and do not require CD36.

Response: We agree that short chain FFA could also be important in this response.

4- The discussion is very short and does not provide much insight into the relevance of the findings and their novel contribution to our current knowledge.

Response: The discussion has now been edited to provide more insight into the relevance of our findings with respect to current knowledge.

Reviewer #3 (Remarks to the Author):

The manuscript by Mistry et al focuses on the role of FFA in HSC functions following infection. The authors suggest that infection facilitates FFA uptake via CD36 and a metabolic switch to beta-oxidation. While, the subject matter is indeed of interest, there are many gaps in this paper that need to be addressed. They are listed below.

Thank you for your comments. We have now addressed the gaps you have highlighted below in your review.

1) The authors claim to have developed a novel assay for FFA uptake (Figure 1C-D). It would be important to describe the principle of this assay in greater detail.

Response: The detail regarding this assay has been included - page 12.

2) There are two Figures 1D. Please correct.

Response: Corrected

3) The FACS plots shown in Figure 1E are difficult to interpret. There seems to be a technical issue with this experiment. C-Kit staining does not seem to work properly and the separation of the LSK population is not convincing (i.e. LSK cells are hardly visible). Furthermore, the CD150/CD48 staining shown does not allow for a confident separation of HSCs. In summary, the LSK and HSC staining needs to be optimised.

Response: The FACS staining has now been optimised and we have shown this in figure 1F. Also where flow is used we have shown representative plots.

4) The data shown in Figure 2A and D display OCR of LSK cells. However, no statistical analyses (P values) are shown. Furthermore, statistical analyses are not offered for results shown in Figure 2C.

Response: Statistical analysis has now been provided for figure 2A in bar charts 2B and 2C. 2D now has stats analysis and so does what was 2C (now 2E) in 2F and 2G.

5) In some cases, etomoxir is referred to as Eto, in other cases as ETX. This needs to be corrected.

Response: This has been corrected

6) Given a possible technical issue with LSK/HSC staining, it would be important that the authors display the LSK/HSC FACS plots and the Ki67 staining.

Response: FACS data is now provided for experiments in Figure 1F, 3H (ki67), Supplementary figures, 4 (Cd36 expression in LSK and HSC), 6 (CD45.1 chimerism in CD45.2 transplanted animals), 8 (myeloid/lymphoid ratios in transplanted animals) and 9 (HSPC engraftment in transplanted animals).

7) When referring to the data shown in Figure 3B-C, they state that HSCs are undergoing expansion. It would be important to also display the total numbers of LSKs and HSCs (e.g. per leg or per bone) to document the actual change in cell numbers.

Response: This has now been provided in figure 3D, 3E and 3J

8) The authors state: "Mice received adoptive transfer of WT (LK con KD) or CPT1A KD (LKCPT1A KD)..." – the exact details of this experiment need to be provided. The knockdown efficiency of CPT1A must to be shown by WB or qPCR.

Response: CPT1A knockdown in the LK is now described in legend of supplementary figure 2C and D. The knockdown efficiency and engraftment of these cells is reported in supplementary figure 2C and 2D.

9) In Figure 4, CD36 KO mice are used. No details about any possible phenotypes are provided. It would be highly appropriate to treat CD36 WT and KO mice with LPS and control and perform detailed analyses of haematopoiesis at different timepoints and bone marrow HSCs and progenitor cell compartments. It would also be appropriate to sort HSCs from these mice and transplant them to test their multilineage reconstitution potentials. It will also be important to analyse the cell cycle profile in HSCs from untreated and treated CD36 WT and KO mice. Multiple functional assays are missing and therefore this study is premature and underdeveloped.

Response: Phenotypes are identical in CD36 WT compared to KO mice. One reason for this may be that we only treated the animals for 16h with LPS. We have now performed the experiments the reviewer suggested and transplanted HSC from CD36^{-/-} from control and LPS treated animals. FACS analysis of myeloid/lymphoid ratios and frequency of GMP, CMP and MEP are reported from these transplants in supplementary figure 7 and 8.

10) Again, Figure 4J shows Seahorse data but no statistical analyses are offered. Are there no significant changes between the groups shown in this figure?

Response: The statistical analysis for this experiment is now provided in supplementary figure 5B

11) The experiments shown in Figure 5A are not introduced in sufficient detail to allow the full appreciation of the results presented here.

Response: This has now been amended

12) The authors state: “These findings provide mechanistic understanding of the interplay between HSC and the bone marrow microenvironment” – where are the data supporting this interplay?

Response: we agree this is misleading and have amended this sentence

REVIEWER COMMENTS

Reviewer #1 (Remarks to the Author):

Based on the comments of the reviewers the authors have added additional data and information to the text to improve the manuscript. However, some questions remain:

- They have added examples of FACS plots but do for example not show the difference between control and LPS plots. For the same HSC stainings there is quite some variation between the different examples of FACS plots as well.
- Regarding the time points of analysis; they refer to the levels of IL-6. However IL6 is not linked to the changes in cell cycle distribution of the HSCs. The response to acute infections is transient with differences in response at different timepoints after the infection. Since their main focus is the cell cycle entry correlated to the FFA uptake, a kinetics of the cell cycle entry and the FFA uptake should be used and compared to identify the best time point for the rest of the analysis.
- They have not answered the question regarding the multiple FFA transporters to be induced upon Sal and LPS treatment (Fig 4A), yet inhibition of a single transporter (CD36) is reducing Ki67 levels back to CTR levels.
- The chimeras in figure 5 were repeated, yet again only wt HSCs into a KO mouse, and not the other way around, which would have been very informative about the stroma and hematopoietic involvement
- I'm not convinced this story is of level of novelty/impact for a publication in Nature Comm.

Reviewer #2 (Remarks to the Author):

The manuscript is improved in clarity by the addition of more experimental details and some new data. The authors modestly improved the discussion, which still does not integrate the findings into what was previously known.

Comments:

The authors speculate that IL6 is inducing adipose lipolysis and FFA uptake by HSCs. However, it is well documented that LPS itself increases lipolysis by adipose tissue via a TLR/ERK1/2 mechanism, e.g. PMID: 28854364. This could be addressed since LPS is being used.

It would be helpful to discuss the evidence that fatty acids regulate the fate of other types of stem cells and that induction of the CD36 gene during HSC maturation could involve histones and chromatin modifications, e.g. PMID: 19128795.

The observation that CD36 is important for upregulation of fatty acid uptake by HSC allowing HSC expansion and facilitating the response to infection is interesting. The novelty of the observation is somewhat limited by the perception that outcome was expected based on previous work by the authors, and by that of others e.g. PMID: 28854364 notably the group of Peter Carmeliet.

The last sentence of the discussion needs to be revised for clarity of meaning.

Response to reviewers

Reviewer #1 (Remarks to the Author):

Based on the comments of the reviewers the authors have added additional data and information to the text to improve the manuscript. However, some questions remain:

- They have added examples of FACS plots but do for example not show the difference between control and LPS plots. For the same HSC stainings there is quite some variation between the different examples of FACS plots as well.

Response: Originally, due to space limitations within figure 1, we chose to use salmonella as a typical illustrative flow plot. As suggested, we have now added LPS treated flow plot for HSC in supplementary figure 1H, as well. Experiments were performed as described and repeated over a period 36 months and all flow data was collected over this period. Moreover, some samples were fixed and permeabilised (figure 4) and others were live cell staining (figure 1). To determine if this step causes variation in FACS plots we took the same bone marrow sample for live cell and fixed and permeabilised. FACS plots below show a clear difference as fix and permeabilise step reduces FSC and SSC staining. We therefore believe this is probably causing the variation between plots in Figure 1 and 4.

- Regarding the time points of analysis; they refer to the levels of IL-6. However IL6 is not linked to the changes in cell cycle distribution of the HSCs. The response to acute infections is transient with differences in response at different timepoints after the infection. Since their main focus is the cell cycle entry correlated to the FFA uptake, a kinetics of the cell cycle entry and the FFA uptake should be used and compared to identify the best time point for the rest of the analysis.

Response: As suggested, we have now performed ki67 staining for multiple time points for both LPS and Salmonella experimental models, to determine an optimum time to perform the FFA uptake assays. The data for these experiments are shown in supplementary figure 1F and G. These data confirm that 16 hours for LPS and 72 hours for Salmonella are appropriate time points to examine FFA uptake.

- They have not answered the question regarding the multiple FFA transporters to be induced upon Sal and LPS treatment (Fig 4A), yet inhibition of a single transporter (CD36) is reducing Ki67 levels back to CTR levels.

Response: To answer this question we have performed additional experiments to examine FATP4 and FABP3 protein expression on HSC in response to LPS and Salmonella. We find that FABP3 expression is elevated in response to LPS, but not salmonella. We also show that FATP4 protein is not elevated in response to LPS or salmonella (supplementary figure 4A and B). To determine if LPS induced FABP3 expression is important in the uptake of LCFA in HSC we used a generic FABP inhibitor. Supplementary figure 4C shows that FABP inhibitor does not block the uptake of LCFA in HSC, whilst the CD36 inhibitor (SSO) does. These new data suggest that CD36 is the prominent FFA transporter in HSC under pathogenic stimuli.

- The chimeras in figure 5 were repeated, yet again only wt HSCs into a KO mouse, and not the other way around, which would have been very informative about the stroma and hematopoietic involvement.

Response: We apologise if we miss-interpreted your comment in the first revision. This stated 'Question here is when the analysis were performed, whether there was a difference in reconstitution of wt vs CD36KO cells and how the observed liver phenotype is linked to CD36 expression in HSCs'.

To address this comment, we demonstrated that wt HSC and CD36 KO could be transplanted into WT animals and that the reconstitution of the bone marrow was similar to control transplants (supplementary figure 7E, 8 and 9).

- I'm not convinced this story is of level of novelty/impact for a publication in Nature Comm.

Response: Thank you for taking time to re-review this manuscript. We respectfully disagree with this point. Previously, we and others have shown that FFA uptake and metabolism is an important aspect of leukemia proliferation (Shafat et al Blood and Ye et al Cell Stem Cell). Here, we show for the first time that normal HSCs use FFA uptake through CD36 to drive B-oxidation and HSC expansion in response to infection. Thanks to your constructive comments, these data provide mechanistic insight of the metabolism of HSCs during pathogenic challenge and we believe this story will be of interest to a wide group of researchers who read Nature Communications.

Reviewer #2 (Remarks to the Author):

The manuscript is improved in clarity by the addition of more experimental details and some new data. The authors modestly improved the discussion, which still does not integrate the findings into what was previously known.

Comments:

The authors speculate that IL6 is inducing adipose lipolysis and FFA uptake by HSCs. However, it is well documented that LPS itself increases lipolysis by adipose tissue via a TLR/ERK1/2 mechanism, e.g. PMID: 28854364. This could be addressed since LPS is being used.

Response: Thank you for your comments: To address this comment we performed an in vivo experiment in which we inhibited IL-6 with a mouse specific antibody to test the role of this cytokine in LPS induced lipolysis. Supplementary figure 1E shows that blocking IL-6 inhibits LPS induced

serum FFA content. This is an interesting finding and something we plan on pursuing in our future research. We have also included a discussion point on this pathway.

It would be helpful to discuss the evidence that fatty acids regulate the fate of other types of stem cells and that induction of the CD36 gene during HSC maturation could involve histones and chromatin modifications, e.g. PMID: 19128795.

Response: We have now included this in the discussion

The observation that CD36 is important for upregulation of fatty acid uptake by HSC allowing HSC expansion and facilitating the response to infection is interesting. The novelty of the observation is somewhat limited by the perception that outcome was expected based on previous work by the authors, and by that of others e.g. PMID: 28854364 notably the group of Peter Carmeliet.

Response: We, like others, develop our hypotheses based on our previous work and that of others. In addition, we extended the fundamental hypothesis in the present work to provide the mechanism of the changes which we observe. Here, we report that uptake of FFA and FAO is important for HSC expansion in response to stress, and not quiescence (as shown in neural stem cells by the group of Peter Carmeliet). Moreover, using mouse KO models and transplantation studies, as well as various inhibitors we provide mechanistic understanding of how FFA are released into the serum following pathogenic challenge to uptake through CD36 in HSC and their use in CPT1a-dependent β -oxidation in response to infection. The importance of FA uptake and FAO in facilitating HSCs entering expansion in response to stress has not been previously studied, hence we believe the results obtained during this study represent an important new finding. Finally, having described this process in detail, we can imagine further work by teams investigating how benign and malignant stem cells in other tissues utilize FFA in response to cellular stress, and also by those interested in the role of adipocytes in the physiologic response to infection in individuals with particular vulnerabilities to infection, for example older people and people with obesity. Accordingly, we think our paper will be of interest to a wide group of researchers who read this journal.

The last sentence of the discussion needs to be revised for clarity of meaning.

Response: We have now revised the last sentence in the discussion.

REVIEWERS' COMMENTS

Reviewer #1 (Remarks to the Author):

The authors have addressed the remaining concerns.

Reviewer #2 (Remarks to the Author):

The authors have added data that enhance mechanistic interpretation of the findings.

-New data link LPS and IL6 to the increase in fatty acids. However the data are limited to the use of IL6 inhibition and the reduction observed with the inhibitor is partial. The statement that IL6 is directly involved in LPS-induced increase in fatty acids needs to be be toned down.

-Other data added regarding the different proteins implicated in fatty acid transport are helpful, despite being limited to the use of inhibitors. Again it is suggested to tone down interpretation.

-The discussion is more interesting and integrates past knowledge. However, the last sentence of the discussion which has been revised still needs revision. In particular the reference to the role of adipocytes needs more context for clarity. Are the authors referring to fatty acid provision by adipocytes?

Response to reviewer

Reviewer #1 (Remarks to the Author):

The authors have addressed the remaining concerns.

Reviewer #2 (Remarks to the Author):

The authors have added data that enhance mechanistic interpretation of the findings.

-New data link LPS and IL6 to the increase in fatty acids. However the data are limited to the use of IL6 inhibition and the reduction observed with the inhibitor is partial. The statement that IL6 is directly involved in LPS-induced increase in fatty acids needs to be be toned down.

Response: we have now toned down this statement to 'suggesting that IL-6 is involved in regulating lipolysis in response to infection.'

-Other data added regarding the different proteins implicated in fatty acid transport are helpful, despite being limited to the use of inhibitors. Again it is suggested to tone down interpretation.

Response: the sentence on page 7 has now been toned down to read 'Taken together these data suggested that CD36 is the trafficking protein regulating FFA flux in this context.'

-The discussion is more interesting and integrates past knowledge. However, the last sentence of the discussion which has been revised still needs revision. In particular the reference to the role of adipocytes needs more context for clarity. Are the authors referring to fatty acid provision by adipocytes?

Response: we have now changed the last sentence in the discussion to refer specifically to 'provision of FFA by adipocytes'